

# Advancing airborne Doppler lidar wind profiling in turbulent boundary layer flow - an LES-based optimization of traditional scanning-beam versus novel fixed-beam measurement systems

Philipp Gasch[1], James Kasic[2], Oliver Maas[3], and Zhien Wang[2]

[1]Institute of Meteorology and Climate Research, Karlsruhe Institute of Technology, Karlsruhe, Germany
[2]Department of Atmospheric and Oceanic Sciences, University of Colorado, Boulder, CO, USA
[3]Institute of Meteorology and Climatology, Leibniz University Hannover, Hannover, Germany

**Correspondence:** Philipp Gasch (philipp.gasch@kit.edu)

**Abstract.**

There is a need for improved wind measurements inside the planetary boundary layer (PBL), including the capability to sample turbulent flow. Airborne Doppler lidar (ADL) provides unique capabilities for spatially resolved and targeted wind measurements in the PBL. However, ADL wind profiling in the PBL is challenging, as turbulence violates the flow homogeneity

assumption used in wind profile retrieval and thereby introduces error in the retrieved wind profiles. As turbulence is a dominant source of error it is necessary to investigate and optimize ADL wind profiling capabilities in turbulent PBL flow.

This study investigates the potential of a novel multiple fixed-beam ADL system design to provide improved wind information in turbulent PBL flow, compared to traditional single scanning-beam ADL systems. To achieve this, an LES-based airborne Doppler lidar simulator presented in Gasch et al. (2020) is employed and extended in this study.

Results show that a multiple fixed-beam system with settings comparable to those of commonly used single scanning-beam systems offers distinct advantages. Advantages include overall reduced wind profile retrieval error due to turbulence and improved spatial representation alongside higher wind profile availability. The study also offers insight into the dependence of the retrieval error on system setup parameters and retrieval parameters for both fixed-beam and scanning-beam systems. When using a fixed-beam system, an order of magnitude higher wind profile resolution appears possible, compared to traditional

scanning systems at comparable retrieval accuracy. Thus, using multiple fixed-beam systems opens the door towards better sampling of turbulent PBL flow.

Overall, the simulator provides a cost-effective tool to investigate and optimize wind profile error characteristics due to turbulence, and to optimize system setup and retrieval strategies for ADL wind profiling in turbulent flow.

## 1   Introduction

Improved wind measurements are critical for advancing our understanding of atmospheric processes and their representation in weather, climate, and pollution models, especially inside the turbulent planetary boundary layer (PBL) (Baker et al., 2014; Geerts et al., 2018). Ground-based Doppler lidars provide vertically resolved insight into PBL mean wind and turbulence, yet





have limited spatial coverage (Weitkamp, 2005). Combining multiple Doppler lidars in a dual-Doppler technique can provide information on the wind vector components oriented in the dual-Doppler plane direction over distances up to approx. 10 km

(e.g. Fernando et al., 2019; Adler et al., 2020). Airborne Doppler Lidar (ADL) offers a unique ability to observe wind and turbulence over much larger areas (Turk et al., 2020), can take measurements over oceans (Chouza et al., 2016) and complex terrain (Weissmann et al., 2005b) and target localized convective systems (Kunz et al., 2022).

Finer scale measurements of both horizontal and vertical winds are required for understanding flow and turbulence in the PBL (Geerts et al., 2018), for example to improve our understanding of flow in complex terrain and severe weather events.

Existing ADL systems provide wind speed profiles by using a single beam, which is directed by a scanner to point at various viewing directions (Weissmann et al., 2005a; De Wekker et al., 2012; Bucci et al., 2018). The wind profile (e.g. the vertically resolved u,v,w components) is then retrieved from the radial velocities obtained at multiple beam directions using an inversion-based approach, assuming flow homogeneity in the retrieval volume (Leon and Vali, 1998). The retrieved wind profiles are assumed to provide an area-averaged representation of the wind inside the retrieval volume, despite the non-uniform

distribution of the radial velocity measurements. The along-track resolution of wind profile retrievals is limited by the time needed for scanning and the aircraft speed. Existing systems typically provide O($2\,\mathrm{km}$) resolution using slow aircraft, capable of operation inside the PBL (De Wekker et al., 2012; Schroeder et al., 2020), and O($10\,\mathrm{km}$) using jet aircraft above the PBL (Witschas et al., 2017). Other ADL systems have used a continuously nadir staring beam to resolve the vertical wind at higher resolution, enabling the retrieval of turbulent properties of the vertical wind (Kiemle et al., 2011; Chouza et al., 2016; Gasch,

2021). However, as a single beam system can either be scanned or stare vertical, area-averaged wind profiles simultaneous to high resolution vertical wind observations have not been available up to date, but would be highly desirable (Witschas et al., 2017; Gasch, 2021).

Due to cost and size reductions of (especially fiber-based) Doppler lidar systems over the recent years, it is now possible to construct an ADL system which does not use a single scanning-beam and instead uses multiple fixed-beams. Using an

appropriate distribution of the fixed-beams, the need for a scanner unit is eliminated. It is expected that a fixed beam setup leads to an improved wind profile retrieval availability and accuracy, as radial measurements at different azimuth angles are available simultaneously, improving the sampling characteristics. Further, high resolution vertical wind observations from a nadir staring beam are available continuously, establishing a link between the horizontal and vertical wind and enabling the retrieval of turbulence properties. Additionally, dual-Doppler retrievals may become possible, as has been done for airborne

Doppler radar (Damiani and Haimov, 2006; Leon et al., 2006).

Two such novel fixed-beam ADL systems are currently under development at the University of Colorado, Boulder (UCB) and the Karlsruhe Institute of Technology (KIT). Both systems aim to provide high resolution measurements inside the PBL for both horizontal and vertical winds and will be installed onboard medium-range turboprop aircraft capable of operation inside the PBL. In order to do so, the systems are designed to contain at least five independent lidar systems.

The present study characterizes and optimizes the wind profiling quality of the envisaged fixed-beam systems in turbulent PBL flow. The focus is put on wind profiling accuracy in a turbulent PBL, as this is an important but challenging measurement task. Turbulence introduces error in the retrieved wind profiles due to the violation of the flow homogeneity assumption



employed in the wind profile retrieval. The error due to turbulence is a dominant source of error in PBL wind profiles measured by ADL (Gasch et al., 2020), as a high level of quality control is possible for other sources of error. For example, the error due to uncertainty in beam pointing directions can be minimized using ground-return based calibration and motion correction schemes (Chouza et al., 2016; Gasch, 2021). In order to investigate the wind profiling error due to turbulence, an LES-based airborne Doppler lidar simulator (ADLS) presented by Gasch et al. (2020, abbreviated as G20 in the following) is employed and extended in this study. ADLS studies have distinct benefits in addition to real-world comparisons. First, the LES input is known and can be used as a reference truth at all locations inside the measurement volume, thus representation errors can be accounted for and investigated. In real-world comparison studies, representation errors complicate or even prevent isolation of wind profile retrieval error due to turbulence. ADL exhibits 3D volume sampling characteristics compared to 1D reference measurements, e.g. dropsonde or aircraft in-situ measurements, thus it is often unclear if observed differences are due to sampling volume differences or ADL retrieval error (Weissmann et al., 2005b; Bucci et al., 2018). Second, the ADLS allows for a flexible setup of system geometries before system production (e.g. number of fixed beams and their orientation), changes which are not easily possible in real-world systems. Overall, ADLS presents a cost-effective tool to investigate and optimize wind profile error characteristics due to turbulence.

The possibility of an LES-based aircraft measurement simulation was devised more than 20 years ago by Schröter et al. (2000) and has since been applied more often (Sühring and Raasch, 2013; Sühring et al., 2019; Petty, 2020). However, the mentioned studies focus on the simulation of in-situ sensor measurements to validate uncertainty estimation methods for aircraft measured turbulent fluxes (Lenschow et al., 1994).

For ADL systems, studies by Reitebuch et al. (2001); Lorsolo et al. (2013); Guimond et al. (2014); Didlake et al. (2015) and Helms et al. (2020) have shown the importance of ADL simulator studies using coarser resolution model output. These studies focused on retrieval errors introduced by measurement system errors, as the coarser resolution models used by them did not represent small-scale turbulence inside the PBL. The finest resolution is used by Helms et al. (2020) with $1\,\mathrm{km}$ model grid spacing, which is still too coarse to represent PBL turbulence. A simulation of an ADL wind profiling system based on high resolution LES wind fields ($O(10\,\mathrm{m})$) was conducted by G20. They show that an LES-based ADL simulation can be used to investigate the wind profile retrieval error introduced by turbulence (due to violation of the flow homogeneity assumption assumed in the retrieval) in the PBL for a commonly used scanning-beam measurement system setup and retrieval strategy. Due to the importance of the wind profile retrieval error due to turbulence, a number of studies exist which investigate its magnitude and characteristics for ground-based systems (Lundquist et al., 2015). Recently, both Rahlves et al. (2022) and Robey and Lundquist (2022) use an LES-based simulator approach to investigate wind profiling error characteristics for ground-based systems. In their studies, the effect of different system setups, e.g. different scan strategies for ground-based scanning-beam lidar systems are investigated.

To our knowledge, a system setup and retrieval strategy optimization study for ADL wind profiling systems in turbulent flow is missing up to date, especially with respect to novel fixed-beam approaches. Therefore, this study investigates the expected measurement quality of the envisaged fixed-beam systems in comparison to scanning-beam systems. The ADLS allows for a measurement system setup and retrieval strategy optimization for both fixed-beam and scanning-beam systems.





The following research questions are answered:

- What benefits does a fixed-beam system provide compared to commonly used scanning-beam systems with respect to wind

profiling retrieval quality in turbulent flow?

- How should the fixed-beam system setup and retrieval strategy be optimized for wind profiling in turbulent flow (e.g. beam elevation and azimuth orientation)?

To answer these question, this study extends and applies the ADLS presented in G20. In the following, Sec. 2 provides an overview of the ADLS and changes therein compared to G20. Sec. 3 compares the wind profile retrieval quality between

scanning- and fixed-beam systems for a commonly used system setup and retrieval strategy. Sec. 4 presents a system setup optimization and Sec. 5 investigates the influence of retrieval settings on wind profile retrieval quality. Finally, Sec. 6 draws conclusions on the results.

## 2   LES-based ADL simulations

To optimize both scanning-beam and fixed-beams systems the ADLS presented in G20 is adapted to allow for flexible combi-

nation of multiple fixed-beam measurements. Compared to G20, the underlying LES data set is extended to a larger domain and longer simulation time in order to generate more reliable statistics. In addition, five LES background wind speeds are used in this study, including two cases in the low wind speed regime, which was shown to be a challenging environment for ADL measurements in G20. Further, the new LES simulations are driven by a higher surface sensible heat flux, which is comparable between all background wind speeds. The higher sensible heat flux generates a more turbulent PBL and thereby increases the

wind profile retrieval error, which is the investigation focus of this study.

### 2.1   New LES set

The LES set is obtained using the PALM model developed at the Leibniz Universität Hannover. The simulations are conducted using PALM version 6.0. The LES set employs a simulation domain size of $23\,030\,\mathrm{m}$ x $17\,270\,\mathrm{m}$ x $2300\,\mathrm{m}$. Vertically, the output is limited to $1500\,\mathrm{m}$. Vertical profiles of the average wind speed, the potential temperature, the kinematic sensible heat

flux and the component-wise wind variances are provided in Fig. B1. An overview of characteristic PBL parameters is provided in Table 1. The LES set is driven with five geostrophic background wind speeds of $u_G = 0, 2, 5, 10$ and $15\,\mathrm{m\,s^{-1}}$ and simulates a dry atmosphere. At the flat surface a constant heating rate of $+0.6\,\mathrm{K\,h^{-1}}$ is prescribed, which results in a kinematic sensible heat flux of approx. $0.13 - 0.17\,\mathrm{K\,m\,s^{-1}}$ (corresponding to a dynamic sensible heat flux of approx. $160 - 210\,\mathrm{W\,m^{-2}}$). In order to dampen the inertial oscillation of the geostrophic wind, a $15\,\mathrm{h}$ pre-run is conducted with a reduced simulation domain.

After a subsequent spin-up time of $4\,\mathrm{h}$ using the full domain (leading to the decay of periodic structures present in the LES), three-dimensional data output began with fully developed turbulence at a temporal resolution of $1\,\mathrm{min}$. In total, $120\,\mathrm{min}$ of data output are available. The convective situation is classified as unstable stratification, organization of the convective structures in the along-wind direction is observed in the LES wind fields for $u_G > 0\,\mathrm{m\,s^{-1}}$ (Salesky et al., 2017). The PBL height is between $1100 - 1300\,\mathrm{m}$, with the entrainment zone extending from $1100 - 1400\,\mathrm{m}$.



**Table 1.** Overview of atmospheric conditions present in the LES. The PBL height $z_i$ is determined from the potential temperature profile.

|  | LES set A | | | | |
| --- | --- | --- | --- | --- | --- |
| Grid spacing in m | 10 | | | | |
| Domain size (length x width x height) in $\text{m}^3$ | 23030 x 17270 x 2300 | | | | |
| Simulation duration in min | 120 | | | | |
| Output temporal resolution in s | 60 | | | | |
| Background wind speed in $\text{m s}^{-1}$ | 0 | 2 | 5 | 10 | 15 |
| Kinematic sensible heat flux $\overline{w'\Theta'}$ in $\text{K m s}^{-1}$ | 0.13 | 0.15 | 0.16 | 0.17 | 0.17 |
| Friction velocity $u_*$ in $\text{m s}^{-1}$ | 0.17 | 0.22 | 0.37 | 0.62 | 0.85 |
| Vertical velocity scale $w_*$ in $\text{m s}^{-1}$ | 1.68 | 1.74 | 1.76 | 1.83 | 1.90 |
| Stability parameter $-z_i/L_0$ | 275 | 157 | 37 | 9.6 | 4.3 |
| Boundary layer height $z_i$ in m | 1100 | 1100 | 1120 | 1200 | 1270 |
| Convective overturning time $\tau^*$ in s | 655 | 632 | 636 | 655 | 668 |

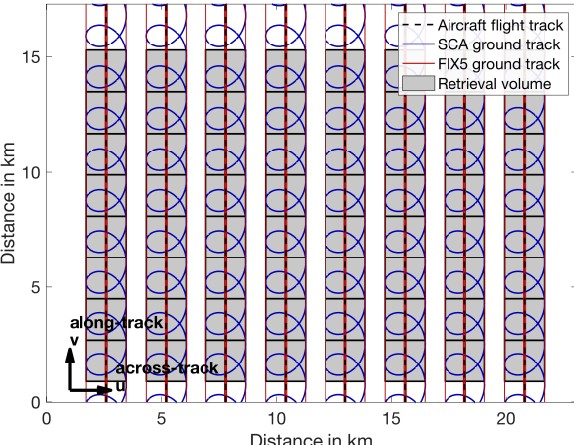

**Figure 1.** Illustration of the checkerboard technique used for wind profile retrieval. Shown are the aircraft transect locations for the crosswind flight direction, the lidar ground tracks for the SCA1 and FIX5 system as well as the extent of the retrieval volumes for a single LES time step (all shown transects are sampled simultaneously). The shown flight transects are repeated every 20 minutes.

## 2.2 Sampling procedure

The sampling strategy is developed based on G20, but using a larger LES simulation domain and longer simulation duration to generate more statistics. Eight parallel flight trajectories traverse the LES domain in a crosswind flight direction with $2600\,\text{m}$ horizontal separation between adjacent trajectories. All profiles are sampled from non-overlapping measurement domains in





space, illustrated in Fig. 1. The eight parallel trajectories are repeated at seven LES time steps with a 20-minute temporal
spacing between each repetition. Thereby, in total 56 transects are obtained for every background wind speed.

To investigate the effect of flight direction, an upwind flight direction is simulated as a second simulation setup. Due to the
reduced LES size in the y-direction, the number of parallel flight trajectories is reduced to six in the upwind case. Thus, for the
upwind case 42 transects are obtained.

The temporal spacing of 20 minutes between transect repetitions ensures independence of repeated transects, given the
convective overturning times $\tau^* \approx 10\,\mathrm{min}$ present in the LES (Tab. 1). While the temporal independence of the retrieved wind
profiles is thereby ensured, it should be noted that correlation (e.g. due to spatial correlation between neighboring profiles)
does not influence the level of the retrieval error discussed in the results section. In any case, profiling error for individual
wind profiles is vertically correlated due to the vertical coherence of turbulence (in reality and in the simulation). Correlation
between wind profile points influences the certainty with which the magnitude of the retrieval error can be estimated (as it
leads to a reduced effective sample size), not the magnitude of the retrieval error itself.

The five LES wind fields are frozen-in-time while the aircraft flies through the LES domain, in-line with the approaches
used by Petty (2020) (in his study only a single LES time step is available) and Helms et al. (2020) (using coarser resolution
model output). The frozen-in-time sampling approach differs from the time-varying approach used in G20, as a much larger
LES domain is used. Due to the much larger LES domain and longer simulation time used here, storing the LES output with
$1\,\mathrm{s}$ temporal resolution is computationally not feasible up to date.

Sampling frozen-in-time LES wind fields relies on the assumption of Taylor's hypothesis of 'frozen turbulence'. In-line
with Petty (2020), based on Lenschow and Stankov (1986), this assumption is valid if $l_w < V * \tau_w$, where $V$ is the aircraft
speed, $\tau_w$ is the temporal auto-correlation time and $l_w$ is the spatial auto-correlation distance of the wind field. In our case,
$V = 100\,\mathrm{m\,s^{-1}}$ and $l_w < 300\,\mathrm{m}$ for all altitudes and wind speed cases in the crosswind direction ($l_w < 500\,\mathrm{m}$ in the along-
wind direction), therefore, $\tau_w > 3\,\mathrm{s}$ is required ($\tau_w > 5\,\mathrm{s}$ in the along-wind direction), which is unproblematic for the PBL
investigated here.

When using a frozen-in-time wind field during sampling, the aircraft trajectory and sampling positions inside the LES must
be calculated differently compared to a time varying wind field. The air mass and turbulent elements contained within are not
advected through the domain during the measurement process. Thereby, the sampling is done at equidistant intervals in LES
space along the flight trajectory. The spacing of the sampling points is calculated using the true aircraft speed (TAS) through
the simple relationship $s = TAS \cdot t$. Consequently, for a given sampling time and TAS, an equal volume of air mass is sampled,
as is done by a real aircraft. However, the aircraft motion due to the wind speed needs to be accounted for during the retrieval
process using a triangle of velocities calculation, because the aircraft track with respect to the ground is influenced by the wind
speed. To illustrate the concept, albeit being unrealistic, consider an aircraft flying at $100\,\mathrm{m\,s^{-1}}$, first down- and then upwind,
with a wind speed of $100\,\mathrm{m\,s^{-1}}$ aligned with the flight direction. In the ground reference frame, the aircraft will have moved
a large distance in the downwind case and not at all in the upwind case. Thus, the number of measurements contained in a
ground-based retrieval volume definition depends on the relationship between aircraft heading and wind vector.



## 2.3 Idealized ADL system setup

**Table 2.** Overview of system setup and retrieval strategy settings. For parameters which are varied the standard values are marked in bold.

| Simulator settings | |
|---|---|
| Parameter | LES set A |
| Background wind case | 5 @ $0, 2, 5, 10, 15\,\mathrm{m\,s^{-1}}$ |
| LES time steps sampled | 7 @ $0, 20, 40, 60, 80, 100, 120$ minutes |
| X-Location of transects flown for every time step | 8 @ $2600, 5200, 7800, 10400, 13000, 15600, 18200, 20800$ m |
| Profiles retrieved per transect | **8**, $16, 24, 32, 64, 120, 240$, depending on along-track averaging distance |
| Aircraft flight altitude | $1500$ m |
| True air speed | $100\,\mathrm{m\,s^{-1}}$ |
| **Beam elevation angle** (from horizontal) | $30°, 40°, 50°, \mathbf{60°}, 65°, 70°, 75°, 80°$ & 1 @ Nadir for fixed-beam systems |
| SCA1 scan rotation speed | $20°\,\mathrm{s^{-1}}$ |
| **Number of fixed-beams** | $3, 4, \mathbf{5}, 6$, symmetric azimuth spacing |
| FIX3, three-beam azimuth directions | 1 @ Nadir & 2 @ $0°, 90°$, variation $0°...180°$ |
| FIX4, four-beam azimuth directions | 1 @ Nadir & 3 @ $0°, 120°, 240°$, variation $0°...180°$ |
| **FIX5, five-beam azimuth directions** | $\mathbf{1}$ **@ Nadir &** $\mathbf{4}$ **@** $\mathbf{0°, 90°, 180°, 270°}$, variation $-6°...80°$ |
| FIX6, six-beam azimuth directions | 1 @ Nadir & 5 @ $0°, 72°, 144°, 216°, 288°$, variation $0°...180°$ |
| Lidar radial velocity gate size | $30$ m |
| Lidar measurement frequency | $10$ Hz |
| **Along-track averaging distance** | $60, 120, 225, 450, 600, 900, 1200, \mathbf{1800}$ m |
| Across-track averaging distance | $5.20, 3.58, 2.52, \mathbf{1.73}, 1.40, 1.09, 0.80, 0.53$ km, depending on beam elevation |
| Vertical retrieval resolution | $30$ m |
| Retrieval altitudes | $100\text{-}1000$ m |

A summary of the system setup parameters is provided in Table 2.

**Aircraft settings**

Similar aircraft settings as used by G20 are simulated in this study. However, the aircraft flight altitude is adjusted to $1500\,\mathrm{m}$ due to the greater PBL height. In addition, a faster aircraft speed of $100\,\mathrm{m\,s^{-1}}$ is used to represent faster medium-range turboprop aircraft.

**Scanning-beam setup**

The scanning-beam system (SCA1) is simulated in accordance with G20 based on commonly used scanning-beam systems applied up to date (De Wekker et al., 2012; Schroeder et al., 2020). Therefore, the beam is scanned with a scan speed of $20°\,\mathrm{s^{-1}}$ at a constant elevation angle. Scan elevation is varied systematically alongside the fixed-beam system to investigate





its influence on retrieval quality. In principle, the scan rotation speed could also be varied. However, in practice the available range of scan speeds is limited. For faster scan speeds, the lag angle becomes noticeable and prevents a high quality return
signal (Ito et al., 2022), as well as a reliable aircraft motion correction (due to the rapidly changing aircraft motion projection). As neither the atmospheric return signal nor the lidar radial velocity measurement process (coherent detection) are simulated using physical models in this study, the degradation of the signal quality with increasing scan speed is not represented in this study. Therefore, a commonly used scan speed setting is used as default and variation is not conducted.

**Fixed-beam setup**

For beam setup optimization, the three adjustable parameters of a fixed-beam system are the number of beams, beam pointing elevation and beam pointing azimuth. This study investigates the influence of these three adjustable parameters on wind profiling quality. To do so, each of the parameters is varied individually with respect to a so-called standard system setup, which enables general conclusions to be drawn.

The standard setup of the fixed-beam system investigated in the following (FIX5) is based on two ADL systems currently
under development at UCB and KIT. These systems will both utilize five lidar systems. While reduced or extended versions are also imaginable, using five beams appears to be a good trade-off between system cost and complexity as well as desired measurement capabilities, detailed in the following.

First, a dedicated nadir beam to observe the vertical wind at highest resolution appears necessary in order to retrieve vertical wind turbulence information along the flight path (Chouza et al., 2016; Strauss et al., 2015; Gasch, 2021). The rest of the beams
can be oriented with a horizontal projection to enable retrieval of the horizontal wind components.

An often used elevation angle for existing scanning-beam systems is $60\,^\circ$ from the horizontal plane ($30\,^\circ$ from zenith), which is thus chosen for the standard system setup.

In order to resolve both horizontal wind components, an obvious choice for the azimuth orientation of the four remaining beams is an equidistant spacing of the beams, resulting in $90\,^\circ$ azimuth angle between them. Two beams oriented forward and
backward along the aircraft axis appear promising, as these beams revisit closely neighboring points as the aircraft passes over. If the aircraft is flying up- or downwind, such a setup may allow for high resolution u,w circulation retrievals on a curtain along the flight path, similar to what has been done using airborne Doppler radar (Damiani and Haimov, 2006; Leon et al., 2006). Orienting the two remaining beams at $90\,^\circ$ azimuth angle results in them pointing to the left and right side of the aircraft flight track. Thus, the across-track wind component can also be measured continuously.

To investigate the advantages and drawbacks of the reasoning and five-beam system design outlined here, the number of beams, beam elevation and azimuth orientation are varied systematically as a part of this study in Sec. 3, enabling generalized conclusions on their influence on retrieval quality.





**Lidar measurement simulation**

The lidar simulation is performed similarly here as in G20, but applying slightly different parameters. In anticipation of the
lidars used for the upcoming fixed-beam systems under development, the range resolution is increased to $30\,\mathrm{m}$ (compared to
$72\,\mathrm{m}$ in G20).

Differing from G20, pulse volume averaging is neglected in this study for two reasons. First, the lidar range resolution is on
the order of the LES resolution in this study (as in other existing LES-based simulator studies, e.g. Stawiarski et al., 2013, and
Gasch et al., 2020). Due to the LES grid spacing of $10\,\mathrm{m}$ (corresponding to a resolution of approx. $50\,\mathrm{m}$), fine scale turbulence
below the scale of the range gate volume is not resolved accurately. Second, linked to the above, the across-beam diameter of
the lidar beam, which is $\mathrm{O}(0.1\,\mathrm{m})$ in real-world measurements, has to be enlarged by a factor of 100 in the simulation in order
to obtain LES grid points inside the measurement volume. Considering these scale mismatches, an adequate representation
of the pulse volume averaging process cannot be obtained in LES-based Doppler lidar simulators. Instead, one could argue
that the LES inherent turbulence smoothing at the finest scales is in itself similar to real-world lidar measurements without
additional pulse volume averaging simulation. The influence of pulse volume averaging on wind profiling retrieval quality is
expected to be marginal due to the averaging involved in the retrieval.

In line with G20, because an ideal measurement system is assumed, neither the atmospheric return signal nor the lidar radial
velocity measurement process (coherent detection) are simulated using physical models. Random radial velocity noise present
in real world measurements is typically of manageable magnitude and becomes negligible considering the averaging inherent
to the wind profile retrieval.

## 2.4  Retrieval strategy

Besides the system setup, the retrieval strategy can also be adapted in real-world measurements. The vertical profile resolution
usually is chosen to correspond to the range resolution of the lidar. A variable retrieval parameter in real-world measurements
is the along-track averaging distance. The along-track averaging distance describes the distance over which measured radial
velocities are considered to retrieve an individual wind profile through the inversion process. For scanning-beam systems an
often used standard setting is the distance covered during one scan revolution (not accounting for the displacement of the
aircraft due to wind discussed above). This approach is also employed here for the standard retrieval strategy and results in an
along-track averaging distance of $1800\,\mathrm{m}$ ($18\,\mathrm{s}$ scan revolution time at $100\,\mathrm{m\,s^{-1}}$ TAS), as displayed in Fig. 1. The along-track
averaging distance is varied between $60\,\mathrm{m}$ and $1800\,\mathrm{m}$ for retrieval strategy optimization, both for the SCA1 and FIX5 system
setups. Vertically, the retrieval is limited between $100\,\mathrm{m}$ and $1000\,\mathrm{m}$ and conducted with $30\,\mathrm{m}$ vertical resolution. The lower
limit excludes near-surface measurements where the LES does not resolve the majority of the turbulent kinetic energy. The
upper limit serves to include only measurements inside the turbulent PBL.

Other parameters, such as retrieving only the u,v components or increasing the vertical averaging interval can also be varied.
However, these parameters have little impact on retrieval error, as shown by ADLS results. For the sake of brevity they are not
included in this study.





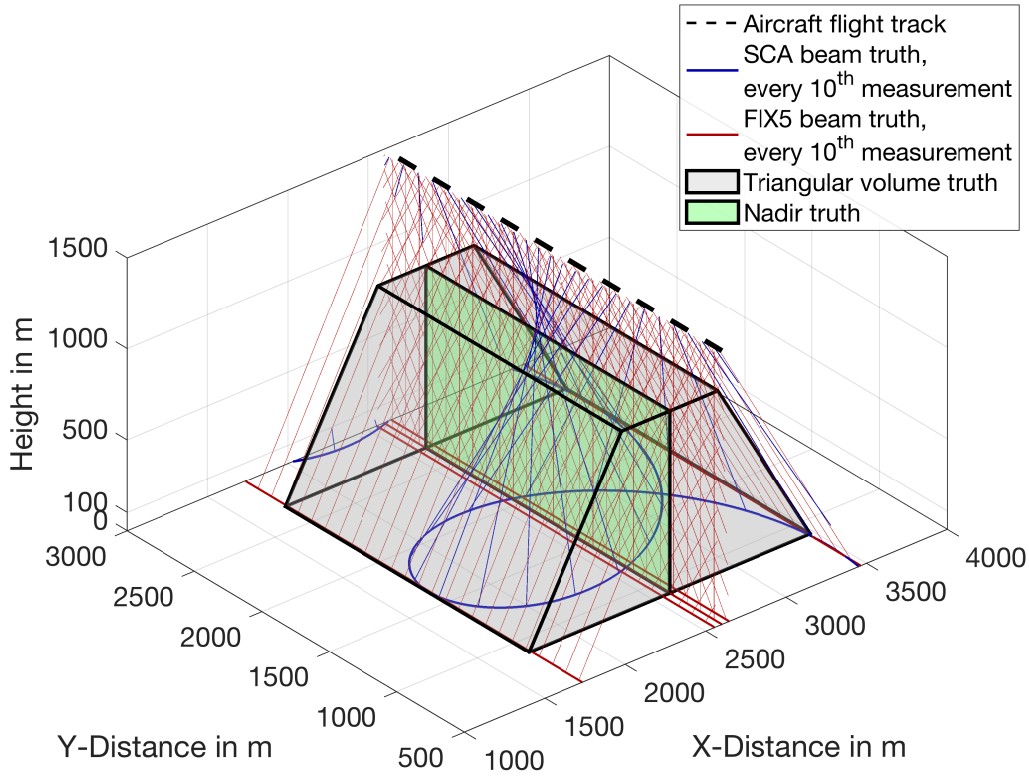

**Figure 2.** Visualization of flight track and lidar measurement geometry including a display of the beam, triangle and nadir truth definition.

## 2.5 Definition of a reference truth and quality metrics

In real-world comparisons, the difference in sampling volumes between the ADL and the reference measurement truth (e.g. from dropsondes) is undesired but cannot be avoided (see Sec. 1).

The ADLS provides a distinct advantage compared to real-world measurements as the input truth used to generate the idealized measurements is known at the point of the measurements, but also everywhere inside the retrieval volume used to retrieve a wind profile (Fig. 2). By defining a beam and triangular volume truth the retrieval error due to turbulence and the spatial representation error can be distinguished. The spatial representation error is especially relevant for fixed-beam systems, due to their strongly non-uniform sampling characteristics, resulting from the along- vs. across-track setup of the beams.

**Beam reference truth**

The so-called beam truth ($u_i^{T,beam}$) is obtained through recording the LES input u,v,w values used to simulate the radial velocity measurements at the location of the range gates. For a single retrieval altitude one can imagine the beam truth as aircraft (one for the SCA1 system, five for the FIX5 system) flying along the positions visited by the lidar beam(s), measuring the u,v,w components without measurement error. Comparison of the ADL retrieved wind profile to the beam truth is done in the studies





by Robey and Lundquist (2022) and the previous ADLS study by G20. A complication arises when comparing scanning- and fixed-beam systems, since their beam truths are different due to the differing spatial sampling characteristics (Fig. 2). Thus, in order to enable a just and reliable comparison, a triangular volume truth is introduced in the following.

**Triangular volume reference truth**

In a real-world scenario, the ADL measurements are assumed to be representative for the retrieval volume, which is determined by the along- and across-track averaging distance (Sec. 2.4). The across-track averaging distance corresponds to the beam elevation and altitude dependent across-track footprint of the ADL (e.g. is decreasing from surface to top) (Fig. 2), thus forming a triangular shaped volume. In the ADLS, the u,v,w LES values inside the retrieval volume are known and can be used as a reference truth, which is termed triangular volume reference truth in the following. The triangular volume truth is equal for systems with comparable system setups and retrieval strategy, further, its comparison with the beam truth allows us to separate retrieval error due to turbulence from the spatial representation error.

Other reference truths, such as a Nadir reference truth (given through the LES values directly below the aircraft position, used by Rahlves et al. (2022) for ground-based lidar simulation) or a square volume reference truth (instead of the triangular shaped reference truth used), are also imaginable and have been implemented in the ADLS. However, as the additional insight into retrieval characteristics provided through these other truths is limited their discussion is not included here for brevity.

**Quality metrics used for evaluation**

The overall retrieval error (the combination of turbulence + spatial representation error) is obtained by comparing the retrieved wind profiles to the triangular volume truth:

$$\text{MAE} = \frac{1}{N}\sum_{i=1}^{N}|u_i^R - u_i^{T,triangle}|. \tag{1}$$

To quantify the spatial representation error $\text{MAE}_{\text{REP}}$ the volume truth is directly compared to the beam truth (e.g. no ADLS retrieval is involved, only LES quantities are used). Thus, the average spatial representation error is obtained as

$$\text{MAE}_{\text{REP}} = \frac{1}{N}\sum_{i=1}^{N}|u_i^{T,beam} - u_i^{T,triangle}|. \tag{2}$$

This procedure can be understood in the following way: The beam truth can be seen as ideal aircraft (e.g. without measurement errors) flying along the positions visited by the lidar beam(s) in the retrieval volume, each recording u,v,w (instead of only a radial velocity). The beam-truth is obtained through simple averaging of the obtained values, hence no retrieval error is present. Despite the absence of a retrieval error, a difference of the beam truth ($u_i^{T,beam}$) compared to the volume truth ($u_i^{T,triangle}$) exists, due to the limited sampling of the retrieval volume by the ideal aircraft (representing the sampling by the lidar beams). This error is the representation error $\text{MAE}_{\text{REP}}$. The $\text{MAE}_{\text{REP}}$ should not be attributed to retrieval error caused by turbulence because it is present even for ideal three component velocity measurements without retrieval.



The retrieval error due to turbulence $\mathrm{MAE_{TURB}}$ hence is the difference between the overall MAE and the $\mathrm{MAE_{REP}}$,

$$\mathrm{MAE_{TURB}} = \mathrm{MAE} - \mathrm{MAE_{REP}}. \tag{3}$$

$\mathrm{MAE_{TURB}}$ is introduced by the limitation of the lidar, which provides only radial velocity measurements (instead of three component velocity measurements) and the subsequent need to perform a retrieval in order to obtain a three component wind vector (for details on the retrieval procedure see G20).

Besides the MAE another useful metric is the bias of the wind profile, as unresolved vertical wind fluctuations on the scale of the measurement volume can results in a biased wind speed retrieval as shown by G20 and Robey and Lundquist (2022):

$$\mathrm{Bias} = \frac{1}{N} \sum_{i=1}^{N} (u_i^R - u_i^{T,triangle}). \tag{4}$$

Additionally, system setup or retrieval options may result in a number of wind profile points not being retrievable. For the scanning-beam system and short along-track averaging distances the retrieval volume may not be adequately explored by radial velocity measurements, in which case condition number (CN) filtering with $CN < 10$ removes wind profile points (see G20). Thus, the normalized number of retrievable wind profile points $\mathrm{N}_n$ is also an important metric. It is calculated as

$$\mathrm{N}_n = \frac{N^R}{N^T}, \tag{5}$$

where $N^R$ is the number of retrieved wind profile points and $N^T$ is the number of theoretically available wind profile points. $N^T$ takes into account changes in the number of wind profile points due to system setup and retrieval strategy changes, e.g. doubling the along-track resolution doubles the number of theoretically available wind profile points.

## 3    Wind profile retrieval quality for standard system setup and retrieval strategy

The simulation setup enables the retrieval of 2240 individual wind profiles (448 for each background wind case), giving 67200 wind profile points for all altitudes (30 wind profile points per wind profile between $100\,\mathrm{m}$ to $1000\,\mathrm{m}$). The 2240 wind profiles are more than what is typically available for comparison in real-world measurements, as co-located validation measurements are difficult and costly to conduct. For example, 33 wind profiles (740 wind profile points) are compared to dropsonde data in Weissmann et al. (2005b), approx. 10 wind profiles to a ground-based wind profiler in De Wekker et al. (2012), a single wind
profile to dropsonde data in Kavaya et al. (2014) and approx. 49 wind profiles (2056 wind profile points) to dropsonde data in Bucci et al. (2018).

### 3.1   Wind speed and direction retrieval quality of SCA1 vs. FIX5 system

Figure 3 shows the wind speed profiling error in a turbulent PBL for both the SCA1 and FIX5 system measuring according to the standard system setup and retrieval strategy specified above and using the triangular volume truth as a reference. A



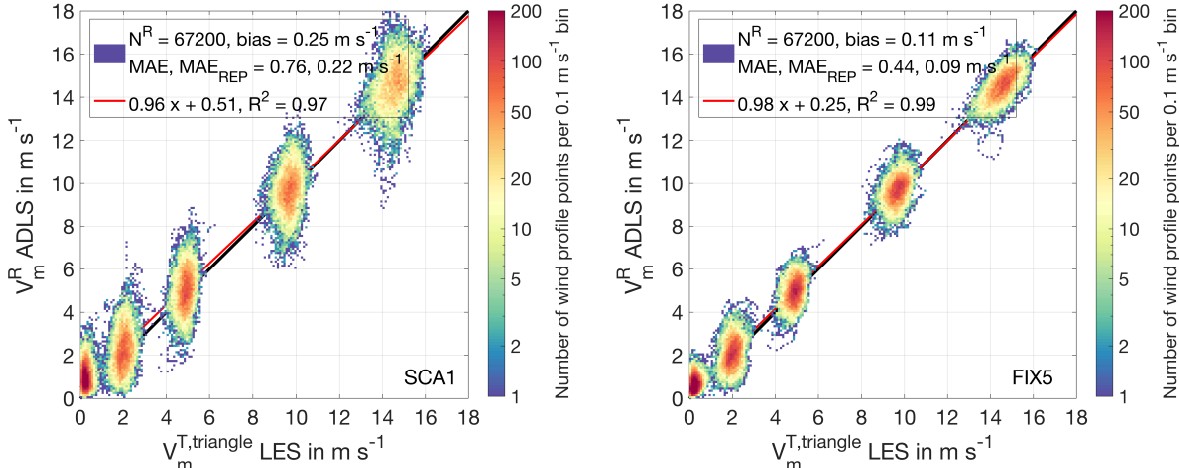

**Figure 3.** Histogram of ADLS retrieved wind speed compared to LES triangular volume truth for the five LES background wind cases using the standard SCA1 and FIX5 system flying in a crosswind direction. Left panel: Results for SCA1 system. Right panel: Results for FIX5 system.

smaller retrieval error of the FIX5 system with an MAE of $0.44\,\mathrm{m\,s^{-1}}$, compared to $0.76\,\mathrm{m\,s^{-1}}$ for the SCA1 system, is evident. Further, the bias of the wind speed retrieval at low wind speeds is reduced for the FIX5 system, as also indicated by the reduced y-intercept (discussed in Sec. 4.1, see also G20, Robey and Lundquist, 2022). Besides the bias, the differences in MAE between the different background wind cases is small. At higher wind speeds, the spread of the input LES triangular truth is larger (e.g. the background wind case retrieval point cloud becomes elongated) due to more strongly reduced wind

speeds inside the PBL. For both systems and all wind speeds, the retrieval error due to turbulence $\mathrm{MAE_{TURB}}$ is much larger than the representation error $\mathrm{MAE_{REP}}$. For the FIX5 system, multiple measurements are conducted simultaneously at different locations in the retrieval volume, resulting in more radial measurements within the same volume size. Thereby, compared to the SCA1 system, the averaging characteristics are improved (lower $\mathrm{MAE_{REP}}$) and the impact of flow homogeneity violations due turbulence is reduced (lower $\mathrm{MAE_{TURB}}$).

The quality of the wind direction retrieval can be evaluated based on Fig. B2. Similar to the wind speed retrieval, the wind direction can also be retrieved with higher accuracy for the FIX5 system compared to the SCA1 system.

   Comparing Figure 3 (SCA1) to the results presented in G20 (scanning-beam system simulation only), an approximately doubled magnitude of the retrieval error is apparent for the SCA1 system (besides the much larger sample size). The increased retrieval error in the present study is due to the more turbulent PBL (the scanning-beam system characteristics are very similar between two studies). The FIX5 system was also included and simulated in the setup of G20 as a test. Similar to the results

shown here, although at overall reduced error levels, the retrieval error of the FIX5 system is roughly half of the SCA1 system (not shown). Therefore, it can be concluded that while turbulence strength influences the absolute magnitude of the retrieval error, the relative differences between the SCA1 and FIX5 systems discussed in the following are applicable independent of turbulence intensity.




## 3.2 Component-wise retrieval quality and sampling characteristics

There is a pronounced dependence of the FIX5 retrieval quality on the relation between the beam orientation with respect to aircraft orientation, flight track direction and wind direction. This dependence is obscured when investigating the wind speed and wind direction retrieval quality, but becomes evident when looking at the retrieval quality of the individual u and v wind components (Fig. B3). The SCA1 system does not show a similarly pronounced directional dependence of the retrieval quality as the FIX5 system due to the more homogeneously distributed measurements inside the retrieval volume.

As a principle, the retrieval from beams sampling in close spatial proximity is better than from beams sampling further apart, due to a better fulfilled flow homogeneity assumption. Besides the distance between flight and measurement altitude, the spatial proximity of beams depends on the system setup, specifically the beam orientation and elevation angle in relation to the flight track direction. For a given setup, the influence on the wind retrieval quality then additionally depends on the beam orientation in relation to the wind direction. A detailed analysis for different beam geometries is provided in Sec. 4.2 but the underlying principle is discussed here already.

**Along- vs. across-track error characteristics of FIX5 system**

For the standard FIX5 system setup investigated here, the forward and aft beams are approximately oriented in along-track direction and thereby sample in close spatial proximity as the aircraft flies over (except for a small aircraft crabbing angle discussed below). For the results discussed so far, the aircraft flies in the crosswind direction, e.g. in the direction of the v-component, as the LES flow is aligned with the u-component (Fig. 1). Due to the beam orientation, flight track and wind direction, the FIX5 system thereby samples the v-component through the forward and aft beams in close spatial proximity. Thus, because the flow homogeneity assumption is better fulfilled, the v-component retrieval quality is higher (Fig. B3). The u-component is retrieved from the side-pointing beams, which are spatially separated (in across-track direction). The spatial distance leads to a less fulfilled flow homogeneity assumption and thereby greater retrieval error in the across-track direction.

Because the LES flow is oriented in u-direction, the retrieval quality of the u-component dominates the wind speed retrieval quality, whereas the retrieval quality of the v-component dominates the wind direction retrieval quality.

For flights in upwind (u) direction the discussed retrieval quality characteristics of the FIX5 system are equally valid, when viewing them with respect to the along-track vs. across-track component, but switched with respect to the u- and v-component. In this case, the u-component (along-track) is resolved by the forward and aft beams with highest retrieval quality, whereas the v-component (across-track) is retrieved from the side-pointing beams, which are spatially separated (Fig. B4).

Stated differently, the retrieval quality of the wind components depends on the choice of flight direction, due to the difference in along- versus across-track retrieval quality for the investigated FIX5 system. However, the results obtained from cross- vs upwind flights are interchangeable, if the discussion is conducted with respect to along-track vs. across-track wind components. Therefore, the following retrieval quality analysis is done with a focus on the crosswind flight direction and discussed separately for the u and v component, which are referred to as along-track component (v) vs. across-track component (u).





**Alignment of flight track and beam orientation**

Additional effects discussed below occur if the flight track and the beam orientation do not align. Such misalignment occurs when the beams are installed differently (e.g. with a different azimuthal orientation) or when the aircraft is crabbing. Crabbing denotes a difference between the aircraft flight track direction and the aircraft heading (nose orientation) due to wind. Crabbing occurs for crosswind flights towards a fixed ground reference, again motivating the the crosswind flight direction as a default for the analysis. The effects of a differing azimuthal system orientation or crabbing are discussed in Sec. 4.2. Further effects can be introduced if the turbulence intensity between the sampled wind components differs (as it does in the LES, see Fig. B1). Increased turbulence intensity in one component degrades the retrieval quality of the affected component, but it does not alter the generalized findings on the effect of system setup and retrieval strategy investigated in the following.

## 3.3 Vertical characteristics

The vertical distribution of the wind profile retrieval error is analyzed by sub-sampling the full distribution shown in Fig. 3 and calculating vertically resolved quality metrics ($MAE_{REP}$, MAE, $N_n$, Bias introduced in Sec.2.5). Results are shown in Fig. 4 for the crosswind and Fig. B5 for the upwind flight direction.

Besides the previously discussed difference in retrieval error for the across-track (u) vs. along-track (v) component, the vertical distribution of wind profile error mirrors that of the vertical wind variance in the LES (Fig. B1). The MAE is largest in the middle of the PBL, where up- and downdrafts have maximum intensity. Towards the ground, a reduction in wind profiling error is observable for all background wind cases. The LES variance profiles of the across-track (u) and along-track (v) component reach their maximum closer to the surface (Fig. B1), but this is not reflected in the component-wise retrieval error. Thereby, the vertical distribution provides another confirmation that the vertical wind is the main driver of wind profiling error (due its dominant projection into the radial velocity measurements for the standard measurement system setup).

Towards the top of the PBL the wind profile error decreases, alongside the reduction in vertical wind variance. Nevertheless, entrainment and detrainment processes can still cause noticeable retrieval error. Further, the contribution of the $MAE_{REP}$ to the overall MAE increases noticeably towards the top of the PBL. As updrafts in the entrainment and detrainment zone have a less homogeneous distribution, less eddies are sampled. Thereby the lidar sampling distribution becomes more important, which is reflected in the increased contribution of the $MAE_{REP}$.

Two noteworthy profile anomalies are visible and require further discussion, although they are located outside the profile range 100-1000 m used for the overall quality analysis. First the MAE strongly increases above the well-mixed PBL between 1300-1400 m for the v-component (independent of flight direction) in the $15\,\mathrm{m\,s^{-1}}$ background wind case. A weaker anomaly is also noticeable for the $10\,\mathrm{m\,s^{-1}}$ case. Second, there is a strong increase of the MAE for the u-component in the surface layer below approximately $50\,\mathrm{m}$ (also independent of flight direction).

Both anomalies are solely attributable to an increased $MAE_{REP}$. Both anomalies are therefore not caused by an increased lidar retrieval error due to turbulence, the $MAE_{TURB}$ remains unaffected at these altitudes. Instead, they occur due to the sampling strategy, e.g. they also occur for ideal measurement systems not-requiring a retrieval (see Sec. 2.5).



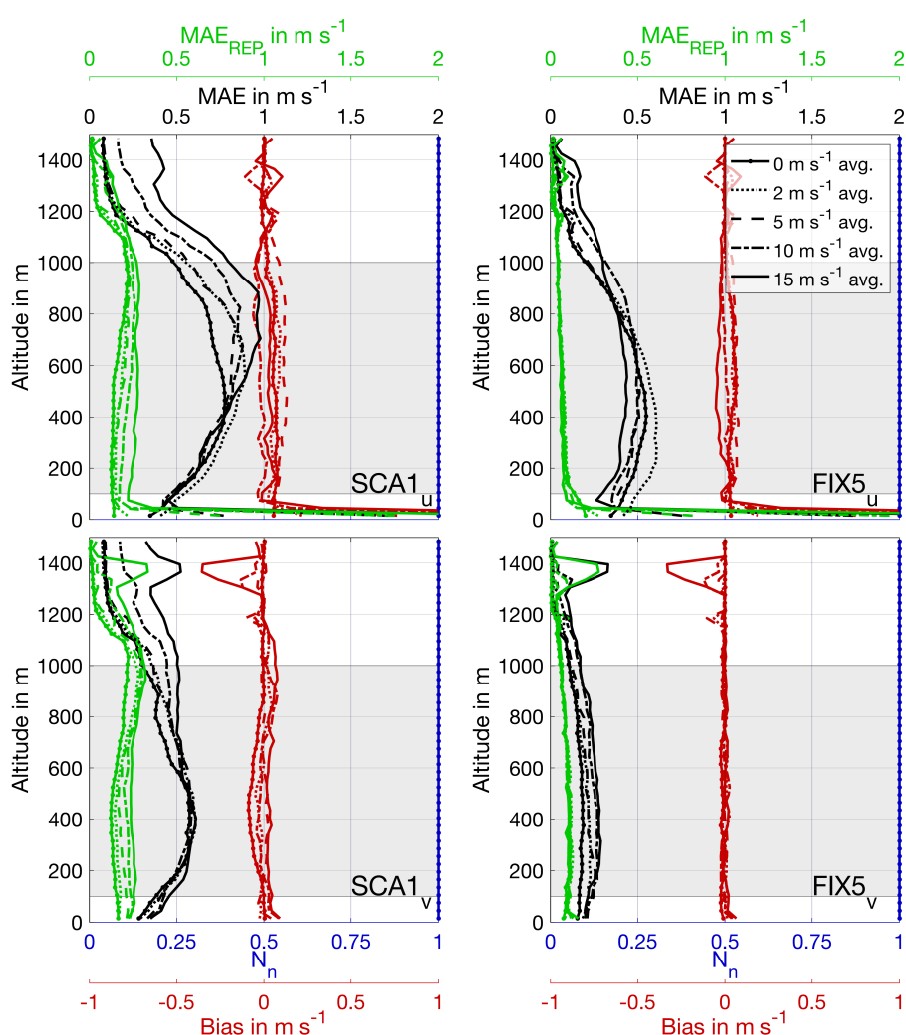

**Figure 4.** Retrieval quality parameters as a function of altitude for the standard SCA1 and FIX5 system flying in a crosswind direction. Left panels: SCA1 system. Right panels: FIX5 system. Top panels: Across-track (u) component. Lower panels: Along-track (v) component. Displayed are the quality metrics $MAE_{REP}$, MAE, $N_n$ and Bias described in Sec. 2.5. The grey area illustrates the vertical section considered for overall quality analysis, system setup optimization and retrieval strategy.





The upper anomaly is explained by a strongly heterogeneous entrainment and detrainment zone above the PBL for the $15\,\mathrm{m\,s^{-1}}$ background wind case. In this region the interplay of updrafts and gravity waves creates a few strong but isolated plume-like structures, which are also detectable in the LES variance profiles in Fig. B1 for v and w. Due to their small spatial scale but strong intensity these structures cannot be adequately sampled using only five measurement locations (of which three are co-located), thus resulting in the increased $\mathrm{MAE_{REP}}$. As the sampling occurs from a strongly non-uniform distribution, the

few strong but isolated plume-like structures also result in a Bias.

The near-surface anomaly occurs in a region where the LES values should be treated with caution, as turbulent scales are not adequately resolved by the LES in this region (see the $E_{GS}/E_{SGS}$ ratio in Fig. B1). For this anomaly, the $\mathrm{MAE_{REP}}$ increases for the u-component with increasing background wind speed. The error is due to the strong non-linear wind shear in the lowest model layers. In regions of strong wind shear the lidar spatial resolution ($30\,\mathrm{m}$ in this study) and pulse volume averaging introduced thereby become important and can introduce error (see Robey and Lundquist, 2022, for a discussion on this). In the

simulations here, the non-linear shear results in a systematic difference between the volume truth and the beam truth (i.e. the observable $\mathrm{MAE_{REP}}$ and bias), because the beam truth is sampled at the geometric center of the volume. Because pulse volume averaging effects are not considered in this study, and due to the questionable representation of turbulence by the LES in the lowest model layers, altitudes below $100\,\mathrm{m}$ are not used in the overall quality analysis (they affect scanning- and fixed-beam

systems equally). The near-surface measurements also require special treatment and investigation in real-world measurements, due to the ground return signal interfering with the measured atmospheric return.

## 4    System setup optimization

It is desirable to obtain a better handle on the wind profiling error in turbulent flow conditions to estimate its effect as well as reduce its impact. Therefore, a number of different system configurations are analyzed systematically in the following and

it is investigated to what extent wind profiling error can be reduced through an appropriate system setup. To do so, the wind profiling quality of a SCA1 system is compared to that of a FIX5 system for different beam elevation angles. Fixed-beam systems offer two additional configuration parameters, therefore, the influence of the azimuthal orientation is investigated as well as the number of beams used in the retrieval.

### 4.1    Beam elevation

An important question for system optimization is at which elevation angle the non-nadir pointing fixed beams should be mounted. The elevation angle can also be varied for scanning systems, thus allowing for a comparison of the SCA1 and FIX5 systems.

The beam elevation angle (measured from the horizontal) is varied between $30°$ and $80°$. Results are provided in Fig. 5 for the crosswind flight direction and in Fig. B7 for the upwind flight direction. For more shallow beam elevation angles (closer

beam the horizontal) the lidar beam covers a larger across-track distance. Consequently, the across-track averaging distance for the triangular volume truth is adjusted, resulting in a larger measurement footprint. For steeper beam elevation angles the lidar





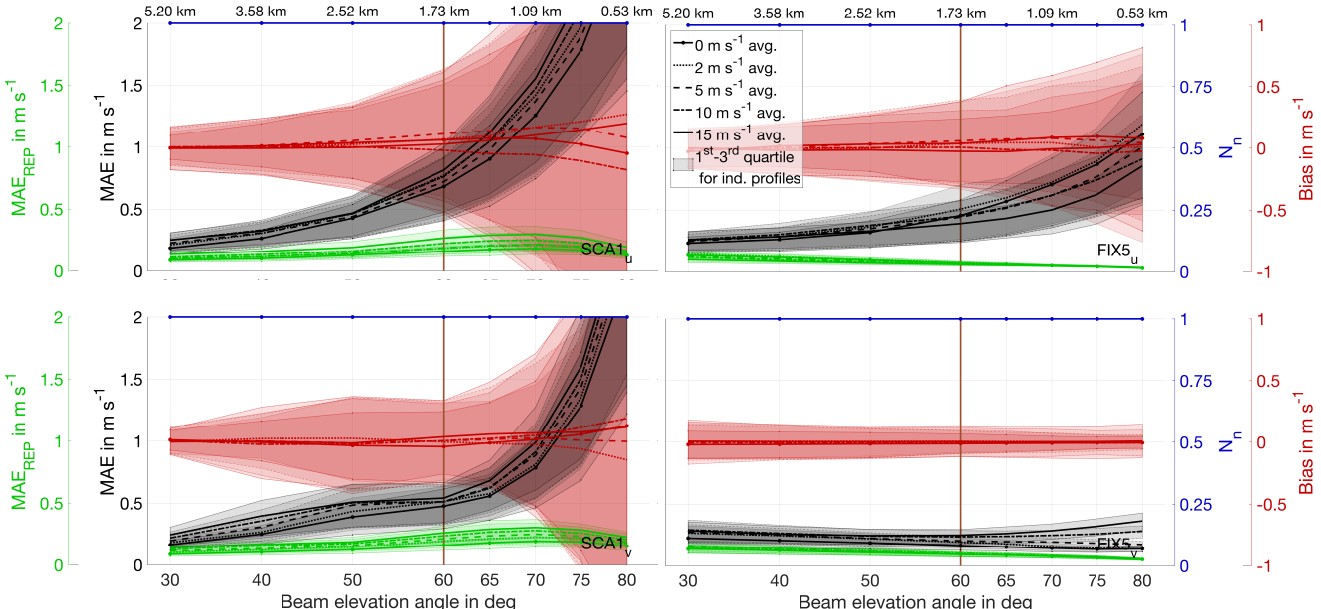

**Figure 5.** Retrieval quality parameters as a function of beam elevation angle for the standard SCA1 and FIX5 system flying in crosswind direction. Left panels: SCA1 system. Right panels: FIX5 system. Top panels: Across-track (u) component. Lower panels: Along-track (v) component. Displayed parameters are separated by background wind speed. Displayed are the mean representation error ($MAE_{REP}$), mean absolute retrieval error (MAE), normalized number of retrieved profile points ($N_n$) and average systematic deviation (Bias). Lines represent average values obtained using all wind profiles, the shaded area indicates the interquartile range for individual profiles. The across-track averaging distance for the beam elevation angles is given on top.

beam covers a smaller across-track distance, resulting in a more confined measurement footprint. The along-track averaging distance is kept constant at the standard value of $1800\,\mathrm{m}$ for all setups.

**Across-track component (u)**

Both SCA1 and FIX5 systems show an increase in MAE at steeper beam elevation angles for the across-track (u) component (Fig. 5 $SCA1_u$, $FIX5_u$). However, the increase in MAE is much stronger for the SCA1 system compared to the FIX5 system. For this component, a FIX5 system measuring at 80° elevation exhibits error levels present in the SCA1 system already at 65°.

In contrast to the MAE, the $MAE_{REP}$ exhibits only slight variation with changing beam elevation. Although it is small overall, the FIX5 system exhibits lower error levels than the SCA1 system, since the retrieval volume is better explored by

the five beams measuring simultaneously. Due to its almost constant magnitude, the $MAE_{REP}$ contributes approximately 50 % of the overall error level at shallow elevation angles but less than 25 % at steep elevation angles beyond 60°. Extending this argument, it is clear that for steeper elevation angles turbulence is the main driver of wind profiling error. Although a slight





decrease in MAE$_{\text{REP}}$ is visible for the FIX5 system at steep elevation angles (due to the more co-located measurements), this decrease does in no way offset the strong increase in retrieval error due to turbulence.

Similar to the MAE, the bias of the individual profiles increases for steep elevations, but remains around $0\,\text{m s}^{-1}$ if averaged. Again, the increase in bias is noticeably larger for the SCA1 system compared to the FIX5 system.

**Along-track component (v)**

An even stronger difference in retrieval quality between SCA1 and FIX5 system is visible for the along-track (v) component (Fig. 5 $\text{SCA1}_{\text{v}}, \text{FIX5}_{\text{v}}$). Whereas the SCA1 system shows again a steep increase in error levels with steeper beam elevation, 440 the FIX5 system does not show such an increase and shows strongly reduced error levels. Due to the co-location of the forward and aft beams for the investigated setup, the along-track (v) component can be retrieved with small MAE and bias even when using steep elevation angles.

The MAE$_{\text{REP}}$ is almost constant with elevation angle. Due to the lower error levels of the FIX5 system, the MAE$_{\text{REP}}$ contributes approximately 50 % to the overall MAE at all elevations.

**Discussion**

Results show that the beam elevation angle has a strong influence on wind profiling quality. Due to the better spatial sampling characteristics, the FIX5 system exhibits generally lower error levels (both for MAE and MAE$_{\text{REP}}$) than the SCA1 system. At steep beam elevation angles the vertical wind exhibits greater influence but the flow homogeneity assumption is not necessarily fulfilled better, as turbulent eddies exist also at small scales. The spectral decay of turbulence mandates reduced variance 450 intensity at smaller scales, however, due to the magnified impact of the vertical wind, this reduction in variance intensity does not result in an improved retrieval. A noticeable exemption is the retrieval of the along-track (v) component by a FIX5 system. In this case, the probed volumes are sufficiently close to fulfill the homogeneity assumption and the increase in error with steeper beam elevation is limited. The next section answers the question on how close is close enough and which other factors besides co-location (or separation distance) are important.

It is important to note that the increasing error levels in the across-track (u) and along-track (v) component can result in a biased wind speed retrieval (Fig. B6). This bias is a result of a non-linear mapping of the wind components into the wind speed and the strong influence of vertical wind perturbations on the scale of the retrieval volume, as discussed in G20 and Robey and Lundquist (2022). In agreement with the theoretical discussion in Robey and Lundquist (2022) the bias appears for the $0\,\text{m s}^{-1}$ wind speed case at all elevations angles and also becomes noticeable for higher wind speeds if beam elevations steeper than 460 $60°$ are used. As for the other quality metrics, the bias is much reduced for the FIX5 system compared to the SCA1 system.

Results for the LES set used by G20 show a very similar qualitative behaviour of the error characteristics, however at approximately halved error levels for both systems due to the reduced turbulence intensity (not shown).





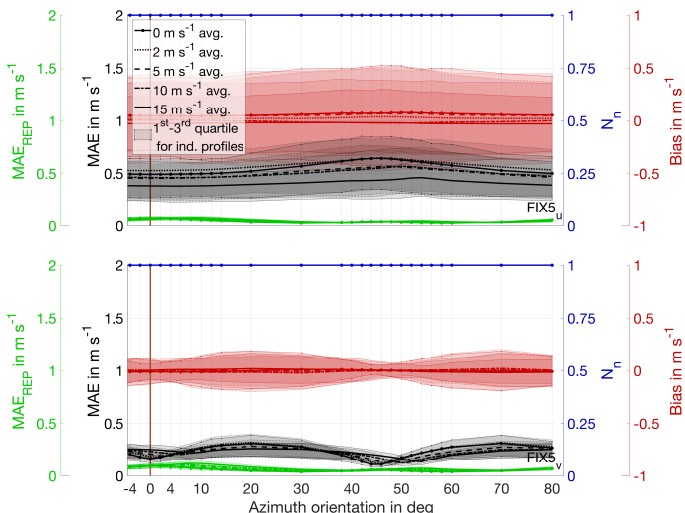

**Figure 6.** Retrieval quality parameters as a function of FIX5 azimuth orientation angle, for the standard FIX5 system flying in crosswind direction. Top panel: Results for across-track (u) component. Lower panel: Results for along-track (v) component. Displayed parameters as in Fig. 5.

## 4.2 Azimuthal orientation of FIX5 system

For a FIX5 system using four horizontal beams as investigated above, the azimuthal orientation of the system can be varied by turning the whole arrangement (but keeping the azimuth spacing between beams at $90°$). On the one hand, such a setup could potentially reduce the spatial representation error of the wind profiles, as the retrieval volume is better explored. For example, at $45°$ the sampling locations of two pairs of beams also align (in this case on the right and left side of the aircraft), leading to another promising system setup option for wind profiling. On the other hand, spatial difference between beams leads to a less fulfilled flow homogeneity assumption.

**Across-track component (u)**

Results for varying the azimuth orientation are presented in Fig. 6 (Fig. B8 for the upwind flight direction). Again a differing behaviour between the across-track (u) versus the along-track (v) component is observed. The MAE of the across-track (u) component retrieval is slightly lower if two dedicated beams sampling the direction are available, e.g. at an azimuth orientation of $0°$ (and $90°$, which is symmetric). For all other orientations the retrieval quality is degraded, which shows that having two dedicated beams sampling the across-track (u) component is better than having four beams sampling a partial projection. The $MAE_{REP}$ is small at all orientations (due to the wide across-track coverage of the beams), but shows shallow maxima when the beams are spatially co-located, as expected.





**Along-track component (v)**

A different picture is observed for the along-track (v) component retrieval, where distinct minima of the MAE are visible
around $0\,°$ and $45\,°$. These azimuth orientations correspond to co-located sampling of beams in the flight direction, resulting in
reduced retrieval error. For the FIX5 system additional alignment with the nadir beam (present at $0\,°$ but not at $45\,°$) appears to
have little additional benefit on the retrieval quality. Again, the $\mathrm{MAE_{REP}}$ increases slightly if beams are aligned, contributing
up to 50 % of the total MAE.

**Effects due to crabbing**

Another noticeable effect is introduced by the crabbing angle of the aircraft. At higher background wind speeds the aircraft is
crabbing more to maintain the same flight track direction in a ground reference frame (see Sec. 3.2). Therefore, the minimum
MAE is reached not at $0\,°$ but at higher angles which correspond to the aircraft crabbing angle. This is due to the mismatch be-
tween the beam orientation (which corresponds to the aircraft orientation) and the flight track direction in ground coordinates.
When the aircraft is crabbing, turning the lidar installation by the crabbing angle results in well-aligned measurements and
highest retrieval quality again. Unfortunately, such a simple offset mechanism cannot be implemented in real-world measure-
ments, since wind shear and thereby advection can vary between measurement levels. Thus, a generalized installation offset
angle is not applicable, even if the crabbing angle is known.

**Effects due to advection on co-located measurements from different beams**

Systematically changing the azimuthal orientation of the installation also provides an estimate of the effect that horizontal ad-
vection between subsequent measurements can have on measurement quality. Greater spatial separation between measurements
from different beams (due to the changing azimuthal orientation) is comparable to the effect that can be caused by advection in
real-world measurements, but is neglected in this study. In real-world measurements the forward and aft beams may measure
at the same geographic location, nevertheless horizontal advection during the time elapsed between measurements can result
in different airmass locations being sensed. The effect of advection can be highly variable in real-world measurements, as it
depends on the crabbing angle and the distance between flight altitude and measurement height (determining beam separation),
as well as the wind speed and direction profile (determining advection at the measurement level).

The frozen-in-time wind field allows for advection to be neglected, thereby the separation effects between subsequent mea-
surements can be systematically investigated (indifferent to whether the separation is caused by actual geographic mismatch of
measurements or due to airmass advection between measurements). Therefore, the results discussed above are also applicable if
the spatial separation is not caused by geographic distance between beams but by advection between subsequent measurements.

## 4.3 Error characteristics of FIX3, FIX4, FIX6 systems

The five beams used by the UCB and KIT systems under construction are not mandatory. Therefore, other options investigated
for comparison in this study (see Tab. 2) are a three beam system (FIX3, nadir and two orthogonal beams for horizontal wind),





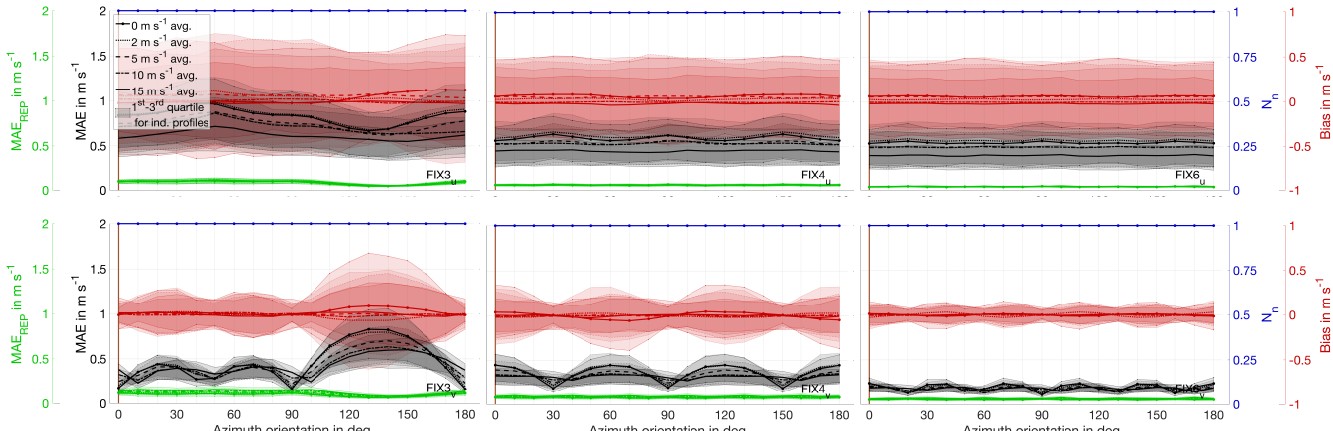

**Figure 7.** Retrieval quality parameters as a function of azimuth system orientation angle for FIX3, FIX4 and FIX6 systems flying in a crosswind flight direction. Left panels: FIX3 system. Middle panels: FIX4 system. Right panels: FIX6 system. Top panels: Across-track (u) component. Lower panels: Along-track (v) component. Displayed parameters as in Fig. 5.

a four beam system (FIX4, nadir and three beams at $120\,^{\circ}$ to each other) and a six-beam system (FIX6, nadir and five beams at
$72\,^{\circ}$ to each other).

Results on the overall and azimuthal error characteristics for the FIX3, FIX4 and FIX6 systems are provided in Fig. 7 for the crosswind flight direction and in Fig. B9 for the upwind flight direction.

Generally, increasing the number of beams results in overall lower MAE and MAE$_{REP}$ levels and reduced azimuthal variability for both the across-track (u) and along-track (v) components. Additionally, as discussed for the FIX5 system already
(Sec. 4.2), beam configurations which result in aligned beams (with each other or with the nadir beam) result in reduced error levels for the along-track (v) component.

The FIX3 system (two non-nadir beams at $90\,^{\circ}$ to each other) possesses beams oriented directly in both across-track (u) and along-track (v) direction (forward and right in aircraft system). Due to the close measurement proximity of forward and nadir beam, the retrieval quality of the along-track (v) component is high (at the cost of a slightly increased representation
error MAE$_{REP}$). However, the across-track (u) component is only covered by a single beam and no co-located vertical wind information, leading to reduced retrieval quality. A secondary minimum occurs in the along-track (v) component at approx. $45\,^{\circ}$ orientation, when the non-nadir beams align spatially. The importance of sampling co-location is also demonstrated by the $135\,^{\circ}$ azimuth orientation results. With respect to the wind component projection into the beams, this setup corresponds to the $45\,^{\circ}$ azimuth orientation setting, however, at $135\,^{\circ}$ all beams sample at spatially different positions. Thus, the retrieval quality
of the along-track (v) component is noticeably degraded, whereas the retrieval quality of the across-track (u) component is slightly improved. Due to the spatial distribution of the measurements both components show a slightly lowered MAE$_{REP}$ at $135\,^{\circ}$.





The retrieval quality of the across-track (u) component is increased when using a FIX4 system (three non-nadir beams at $120°$ to each other), due to two beams sampling projections of the across-track direction. However, in this case the retrieval quality of the along-track (v) component is degraded for an azimuthal orientation of $0°$, due to the missing proximity of the beams sampling the along-track (v) component. As expected, minima in the along-track (v) component MAE occur at $30°$, $90°$ and $150°$ azimuthal orientation, which present beam alignment points of the FIX4 installation.

The retrieval characteristics of the FIX5 system have been discussed in depth already, an increase to six beams (FIX6, five non-nadir beams at $72°$ to each other) yields a further overall decrease in retrieval error, less azimuthal variation and better spatial representation. While the retrieval quality of the across-track (u) component is almost without azimuthal variation, the along-track (v) component still shows weak minima at angles which are beam alignment points of the installation. The FIX6 system does not offer the advantage of the forward and backward beam measuring in alignment with the aircraft flight track. Hence, the minimum in the along-track (v) component retrieval error present in the FIX5 system at $0°$ azimuthal orientation does not exist. Therefore, the slightly reduced azimuthal variation needs to be weighted against the possibility to perform an improved high resolution along-track component retrieval below the aircraft flight track. Overall, a FIX5 system may present a good compromise between retrieval accuracy and cost and complexity in a real-world system.

## 5 Retrieval setting influence - along-track averaging distance

Besides the system setup, the retrieval also contains settings which can be varied. The influence of the along-track averaging distance used to define the wind profile retrieval volume is investigated in the following, again comparing the FIX5 to the SCA1 system. Other parameters, such as retrieving only the u,v components or increasing the vertical averaging interval can also be varied. However, ADLS results show that these parameters have little impact on retrieval error. Therefore, other parameter variations are not included in this study.

The along-track averaging distance is varied between $60\,\text{m}$ - $1800\,\text{m}$ (determining the along-track wind profiling resolution). The along-track distances correspond to the approximate distance covered by the aircraft during $0.0\overline{3}$-1 scan revolution(s).

As expected, varying the along-track averaging distance used for profile retrieval has a noticeable influence on wind profile error, as shown in Fig. 8 for the crosswind flight direction and in Fig. B10 for the upwind flight direction.

**Across-track component (u)**

Both SCA1 and FIX5 system show increasing MAE levels for the across-track (u) component with shorter along-track averaging distance. However, the error levels of the FIX5 system are strongly reduced compared to those of SCA1 system at all distances. The second important difference is the availability of normalized wind profile points at shorter averaging distances. For the SCA1 system, at short averaging distances many retrieval volumes are not adequately covered by measurements, leading to removal of wind profile points by condition number filtering (CN < 10 is applied). For example, at $60\,\text{m}$ along-track averaging distance 2016000 wind profiles points are theoretically retrievable.





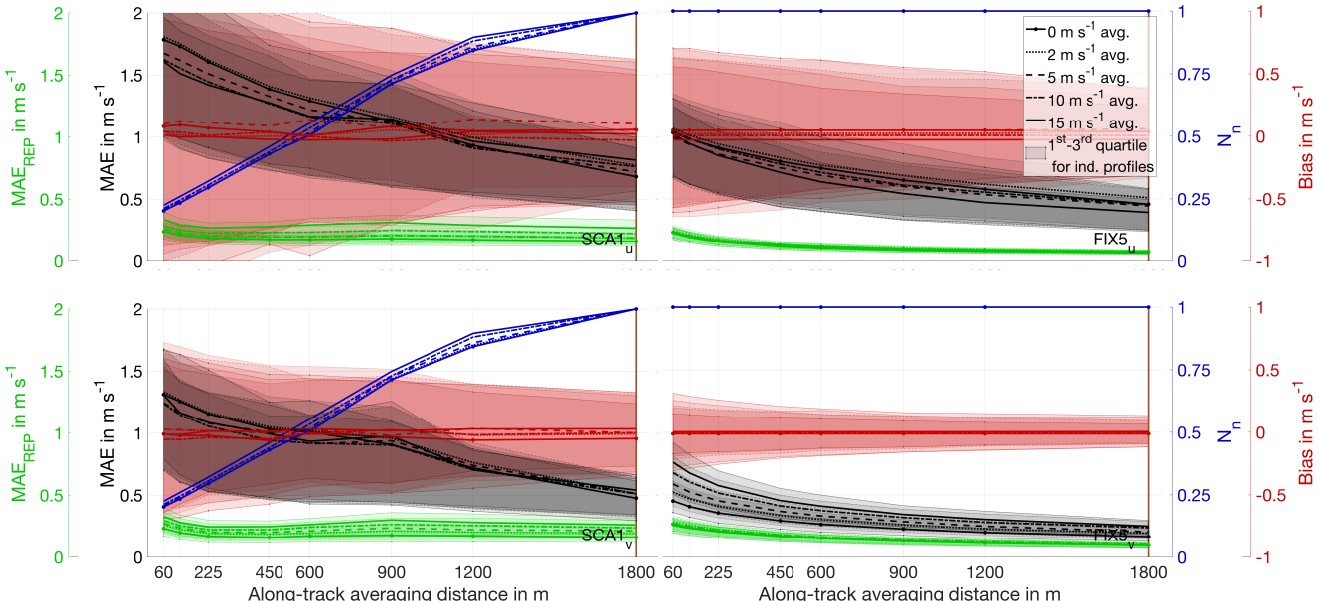

**Figure 8.** Retrieval quality parameters as a function of along-track averaging distance for the standard SCA1 and FIX5 system flying in a crosswind direction. Left panels: SCA1 system. Right panels: FIX5 system. Top panels: Across-track (u) component. Lower panels: Along-track (v) component. Displayed parameters as in Fig. 5.

On the one hand, the SCA1 system can only retrieve approximately 25 % of the theoretically available points (<500000 wind profile points). While this number still presents an increase from the 68400 wind profile points retrievable for $1800\,\mathrm{m}$ averaging distance (although at the price of increased error levels), the majority of wind profile points remains non-retrievable. For the SCA1 system the specific quality and number of wind profile points retrievable depends on the CN filter threshold applied. The influence of this parameter is detailed in Appendix A, which shows that the applied CN < 10 threshold is a reasonable trade-off between the number of retrievable profile points and the quality of the retrieved wind profile points.

On the other hand, the FIX5 system setup allows for retrieval of all theoretically available wind profile points at all averaging distances. This advantage is due to the simultaneous measurement of the five beams, covering all retrieval volumes adequately and thereby having constant CN also for short averaging distances.

**Along-track component (v)**

The strong retrieval quality improvement which can be obtained by using a FIX5 system compared to a SCA1 system is more pronounced for the along-track (v) component. The number of retrievable profile points shows the same behaviour as for the across-track (u) component, as expected. Due to the close spatial co-location of the forward and aft beams in the FIX5 retrieval, the retrieval quality improvement of a FIX5 system compared to SCA1 system is higher for the along-track (v) component. As a consequence of the small footprint of the FIX5 along-track (v) component retrieval, the $\mathrm{MAE_{REP}}$ is slightly elevated



compared to the SCA1 system and increases at shorter averaging distances. It contributes approximately 50 % to the overall
retrieval error. Further, for shorter averaging distances the influence of the increasing crabbing angle of the aircraft at higher
wind speeds becomes noticeable. Larger crabbing angles result in a larger spatial separation of forward and aft beams, leading
to an increased error (see Sec. 4.2).

**Discussion**

Overall, the results demonstrate the potential for strongly improved resolution paired with higher wind profile retrieval quality
when using a FIX5 system compared to a SCA1 system. The expected improvement is on the order of one magnitude with
respect to retrieval resolution and approximately a factor of two with respect to retrieval quality. Especially the retrieval of the
along-track (v) component is strongly improved when using a FIX5 system.

In theory, using a FIX5 ADL allows for further reduced along-track averaging distances down to the spacing of individual
measurements (e.g. 10 m in this study). However, since the retrieval error is of the same magnitude as the spatial variability
at these scales, it is questionable whether such a retrieval is meaningful. Further, distances below 60 m are smaller than the
resolution of the LES. Due to the LES grid spacing of 10 m, turbulence is partially parameterized at these small scales, requiring
further investigation to what extent the LES can be used as a reference truth for such highest resolution retrievals. In principle,
however, it appears possible to also investigate the quality of turbulence retrievals in an ADLS, as the dominant turbulent
scales containing the majority of the turbulent kinetic energy are resolved in the LES. Summarizing, the investigation of ADL
turbulence retrieval quality is beyond the scope of this work but intended for future studies.

## 6 Conclusions

This study compares and optimizes the wind profile retrieval quality of traditional scanning-beam and novel fixed-beam ADL
in turbulent PBL flow. To this end, the ADLS presented in G20 is extended to allow for simulation of multiple fixed-beam ADL
measurements. Further, compared to G20, the underlying LES data set is extended to a larger domain, longer simulation time
and five LES background wind speeds. Beside the strongly increased statistics, the new LES set is also driven by a stronger
surface sensible heat flux, generating a more turbulent PBL.

The extended ADLS is applied in a system setup and retrieval strategy optimization study in preparation of upcoming fixed-
beam ADL systems under development at UCB and KIT. The main drivers influencing wind profile retrieval error due to
turbulence are examined based on a systematic analysis of different system setup and retrieval strategy options. While the
specific level of retrieval error depends on the turbulence conditions present, the ADLS allows conclusions to be drawn on the
behaviour of the retrieval error which hold generally.

Results show that a fixed-beam system with settings comparable to those of commonly used scanning-beam systems offers
distinct advantages. As such, a fixed-beam system offers superior retrieval quality and wind profile availability compared to a
scanning-beam system at all wind speeds. Advantages encompass overall reduced wind profile retrieval error (MAE $0.44\,\mathrm{m\,s^{-1}}$



vs. $0.76\,\mathrm{m\,s}^{-1}$ for standard system), both due to improved spatial representation (lower $\mathrm{MAE_{REP}}$) and reduced retrieval error due to turbulence (lower $\mathrm{MAE_{TURB}}$).

Detailed insight into the retrieval error and its dependence on system setup parameters is gained through a measurement system setup and retrieval strategy optimization. Differing from scanning systems, a fixed-beam system exhibits non-uniform sampling characteristics and retrieves the along-track wind component with higher accuracy than the across-track component.

In spite of this difference, the fixed-beam system resolves both components with higher accuracy than a comparable scanning-beam system. For scanning-beam systems, beam elevation angles steeper than $60°$ are problematic for sampling turbulent wind fields in the PBL, due to the strong influence of the vertical wind on the radial velocity and thereby retrieval error. For beam elevations greater than $60°$, the wind profiling error grows rapidly, both for the across- and along-track wind component, and the wind speed retrieval becomes increasingly biased at low wind speeds. For fixed-beam systems, the increase in retrieval error for

steeper beam elevations is much reduced, especially for the along-track component if sampled from co-located measurements.

Due to the non-uniform sampling geometries, the wind profiling error associated with fixed-beam systems depends on the azimuthal distribution of beams, their relation to aircraft orientation and flight direction, as well as the wind profile itself. Co-location matters, especially for the along-track component: Sampling the same location with multiple beams in close spatial proximity is beneficial for the retrieval quality, as the flow homogeneity assumption is fulfilled better. Additionally, wind

components sampled through dedicated beams are retrieved with less error compared to wind components retrieved from a partial projection into multiple beams. A FIX5 system offers noticeably improved retrieval accuracy compared to systems with fewer beams (FIX3, FIX4), especially at $0°$ azimuthal orientation. Further, the retrieval quality from systems with fewer beams shows an increased directional dependence compared to systems with more beams. A FIX5 system presents a good option to exploit the benefits of sampling in close spatial proximity: The forward and aft staring beams sample the along-track wind

component approximately along the aircraft track as the aircraft passes over. Thus, the along-track component can be sampled with noticeably higher accuracy and higher resolution compared to the across-track component. The co-located sampling also present an advantage of a FIX5 system compared to a FIX6 system (besides reduced cost and complexity), which exhibits less azimuthal variation of retrieval quality otherwise. ADLS results show that already a quite small co-location misalignment of $O(100\,\mathrm{m})$, which often occurs due to crabbing of the aircraft or advection of the wind field between subsequent measurements,

can reduce the retrieval quality advantage of the along-track component noticeably (up to a 50 % increase in MAE).

Retrieval settings also impact wind profiling quality. A fixed-beam system allows for high-resolution retrievals down to very short averaging distances, providing one order of magnitude better wind profile resolution. The continuously available measurements from multiple beams explore retrieval volumes at short along track averaging distance adequately, enabling wind profile retrieval. In contrast, for a scanning-beam system many retrieval volumes are not sufficiently filled, creating impractical

gaps in the retrieval availability. As expected, longer horizontal averaging distances increase retrieval accuracy, whereas at shorter averaging distances the retrieval error is increased. Again, a fixed-beam system shows superior retrieval quality metrics compared to a scanning-beam system for all averaging distances. Typical MAE reductions are up to 50 % for the across-track wind component and even more for the along-track wind component.



Overall, a unique capability of using an ADLS is that design decisions can be made prior to system production and availability. Hence, various potential setups can be evaluated for their measurement quality, a flexibility which is not easily possible in real-world systems. A further advantage of ADLS simulations are their low cost in comparison to real-world aircraft measurements. While validating the wind profile retrieval accuracy of real-world ADL is important nevertheless, the ADLS allows one to estimate expected error characteristics beforehand and without requiring costly flight hours. Further, because the input wind field is known in detail, ADLS simulations provide insight into the spatial representation error, which is difficult to assess when comparing ADL measurements to those from other sensors (e.g. dropsondes, ground-based lidars, other aircraft).

We believe that ADLS studies offer potential beyond what has been presented so far. Future applications of the ADLS could aim to investigate the possibility of simulating and analyzing turbulence retrieval properties of ADL systems, similar to studies conducted for in-situ flux measurements by Schröter et al. (2000) and Sühring and Raasch (2013).

*Code availability.* The MatLab code used for the simulations is available from the first author upon request.

*Author contributions.* PG and JK developed the fixed-beam extension of the ADL simulator presented in G20. OM conducted the LES-runs used in the present study. PG conducted the ADL simulations, prepared the results and wrote the first draft of the manuscript in joint discussion with JK. All authors contributed to the revision of the manuscript.

*Competing interests.* The authors declare that no competing interests are present.

*Acknowledgements.* PG and JK would like to acknowledge Connection Young Scientists (ConYS) funding in support of their collaboration.



**Appendix A: Influence of CN filtering threshold on SCA1 system retrieval quality parameters at shortest along-track averaging distance**

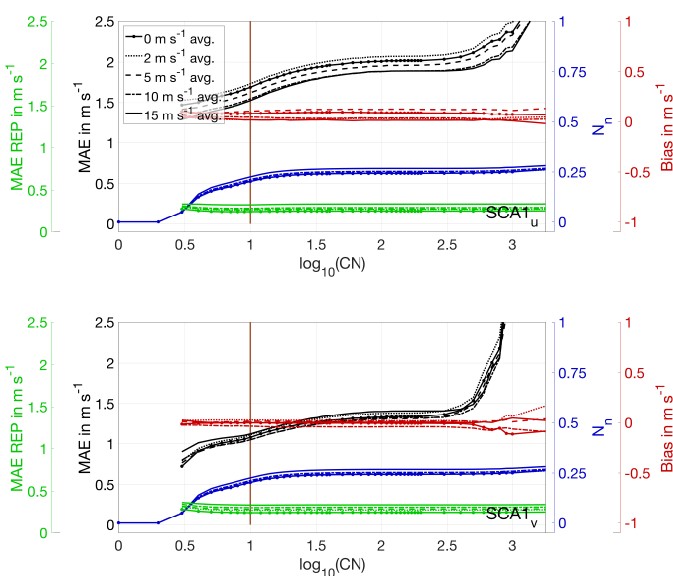

**Figure A1.** Retrieval quality parameters as a function of CN filter threshold applied for the standard SCA1 system flying in a crosswind direction. The along-track averaging distance is $60\,\mathrm{m}$. Displayed parameters as in Fig. 5. Top panel: Across-track (u) component. Lower panel: Along-track (v) component. Displayed parameters as in Fig. 5.

For the SCA1 system the number of retrieved wind profile points and their quality depends on the setting of the CN filtering threshold applied, if averaging distances shorter than one full scan revolution are analysed. In this study CN < 10 is applied, but the specific threshold can be set by the user. Therefore, the effect of using different CN quality filtering thresholds is exemplary displayed in Fig. A1 at shortest along-track averaging distance $60\,\mathrm{m}$. Retrieval of the first wind profile points begins for CN > 3. The number of retrievable points increase strongly until a plateau is reached at $N_n \approx 0.3$ for CN > 10. Alongside the increasing number of retrieved wind profile points the MAE increases. The MAE continues to increase after the plateau in the number of points is reached. Thus the choice of CN < 10 as a threshold is motivated, since it maximizes the number of retrievable points but keeps the MAE low still. For values CN > 300 very few additionally retrieved wind profile points cause a rapid MAE increase to values $> 2\,\mathrm{m\,s^{-1}}$, since retrieval is attempted in volumes that are not adequately explored.





## Appendix B: Additional figures

**Figure B1.** Vertical profiles of the average wind speed, potential temperature, kinematic sensible heat flux, the component-wise wind variance and the grid vs. sub-grid scale turbulent kinetic energy for LES set A. The different background wind cases are color-coded, values are averages over the last 10 minutes of the simulation.



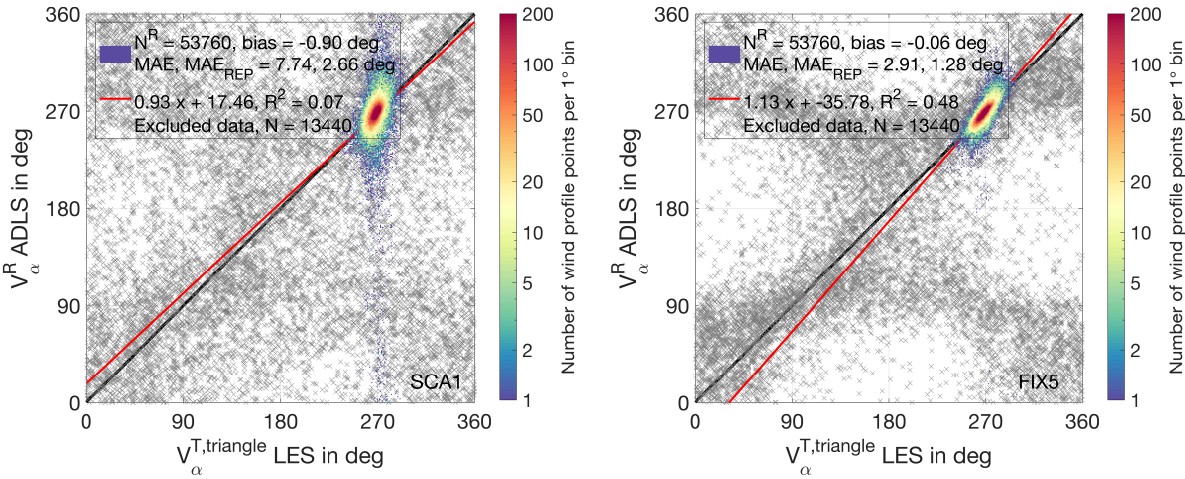

**Figure B2.** Histogram of ADLS retrieved wind direction compared to LES triangular volume input truth for the 2,5,10 and $15\,\mathrm{m\,s^{-1}}$ LES background wind cases and the standard SCA1 and FIX5 system flying in a crosswind direction. The $0\,\mathrm{m\,s^{-1}}$ background wind case is excluded as no meaningful wind direction can be defined for this case. Left panel: Results for SCA1 system. Right panel: Results for FIX5 system.

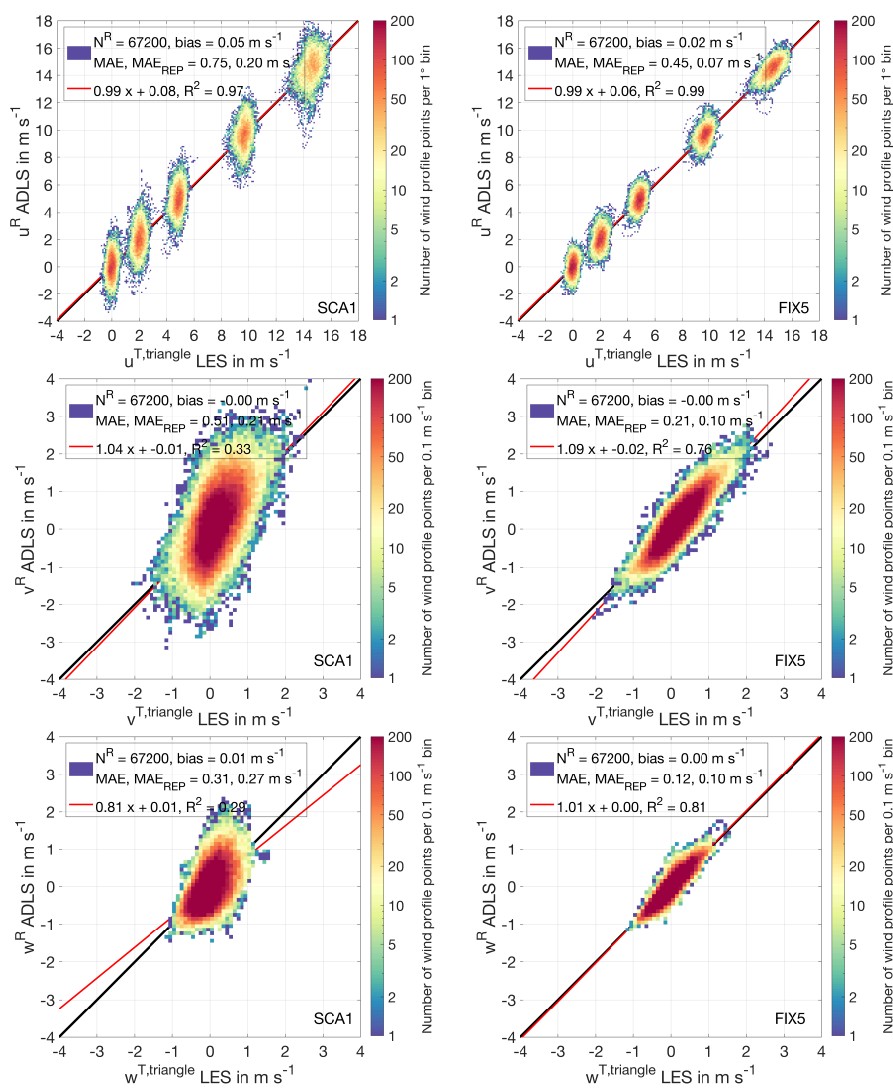

**Figure B3.** Histogram of ADLS retrieved wind speed compared to LES triangular volume input truth for the standard SCA1 and FIX5 system flying in a crosswind direction. Left panels: Results for SCA1 system. Right panels: Results for FIX5 system. Top row: Across-track (u) component. Middle row: Along-track (v) component. Bottom row: Vertical (w) component.



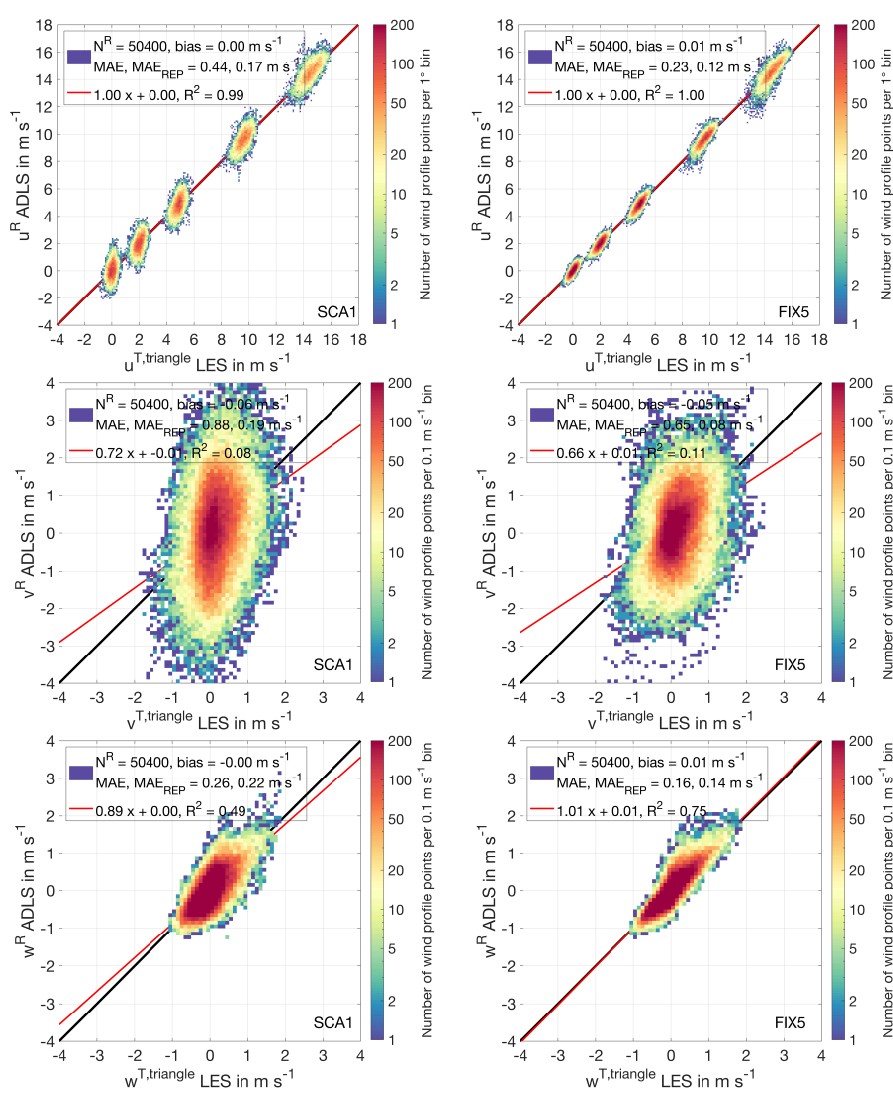

**Figure B4.** Histogram of ADLS retrieved wind speed compared to LES triangular volume input truth for the standard SCA1 and FIX5 system flying in an upwind direction. Left panels: Results for SCA1 system. Right panels: Results for FIX5 system. Top row: Along-track (u) component. Middle row: Across-track (v) component. Bottom row: Vertical (w) component.

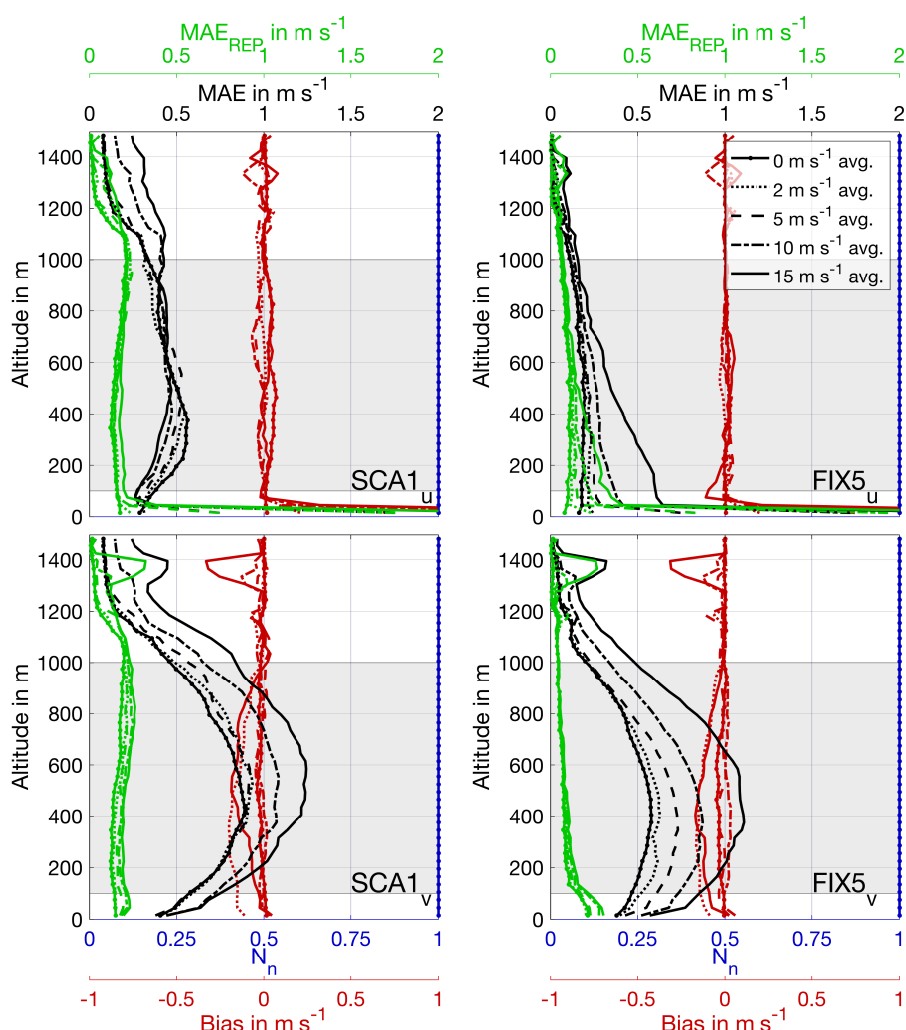

**Figure B5.** Retrieval quality parameters as a function of altitude for the standard SCA1 and FIX5 system flying in a upwind direction. Left panels: SCA1 system. Right panels: FIX5 system. Top panels: Across-track (u) component. Lower panels: Along-track (v) component. Displayed are the quality metrics $MAE_{REP}$, MAE, $N_n$ and Bias introduced in Sec. 2.5. The grey area illustrates the vertical section considered for overall quality analysis, system setup optimization and retrieval strategy.



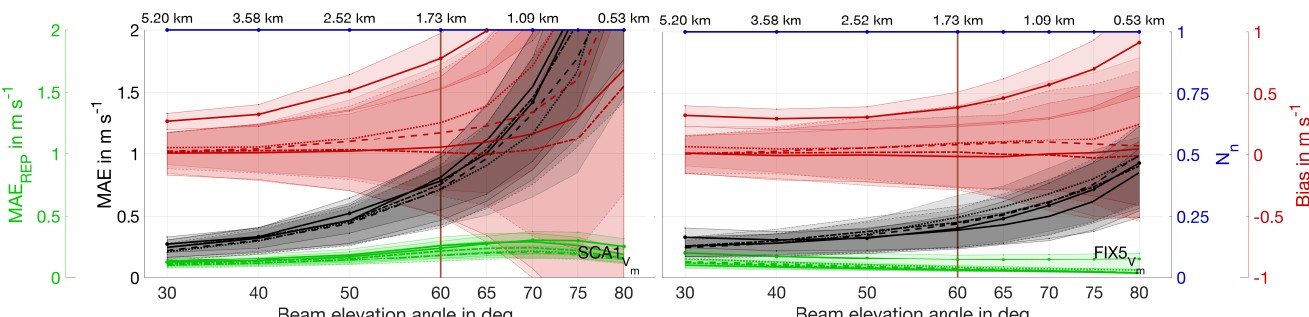

**Figure B6.** Retrieval quality parameters for the wind speed ($V_m$) as a function of beam elevation angle, for the standard SCA1 and FIX5 system flying in a crosswind direction. Left panel: SCA1 system. Right panel: FIX5 system. Displayed parameters as in Fig. 5.

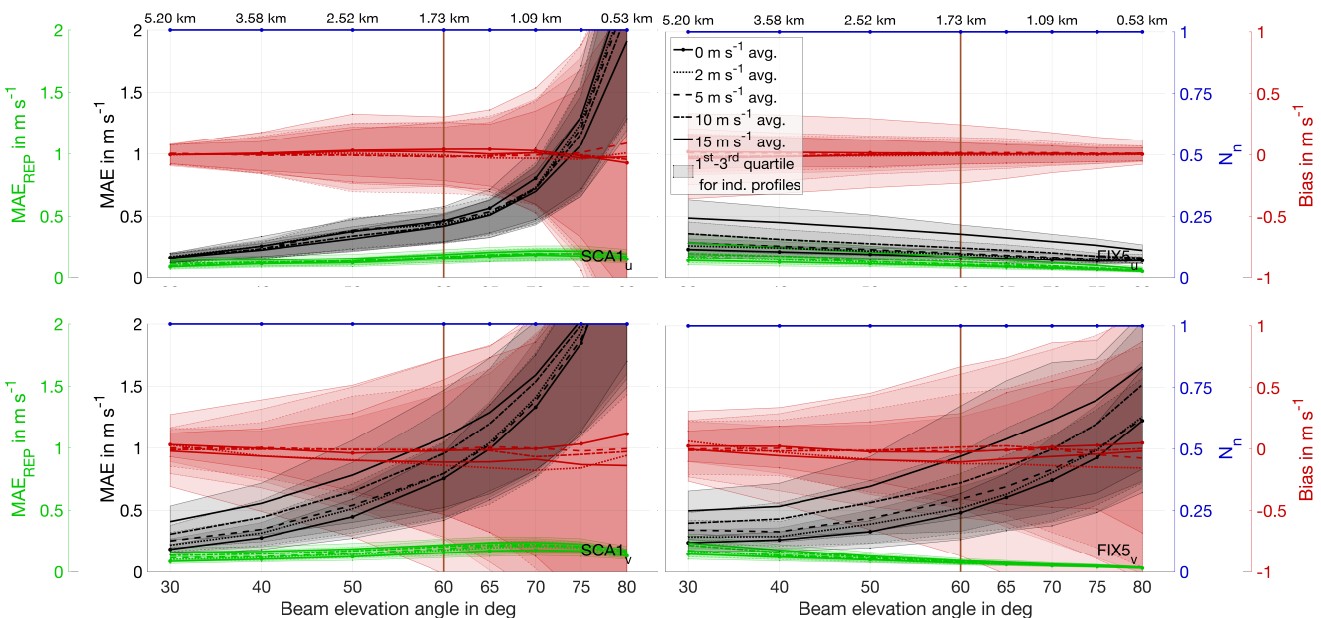

**Figure B7.** Retrieval quality parameters as a function of beam elevation angle, for the standard SCA1 and FIX5 system flying in an upwind direction. Left panels: SCA1 system. Right panels: FIX5 system. Top panels: Along-track (u) component. Lower panels: Across-track (v) component. Displayed parameters as in Fig. 5.



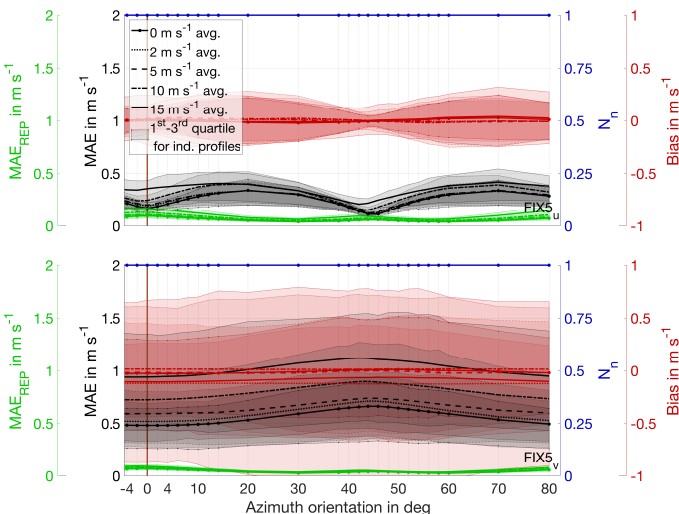

**Figure B8.** Retrieval quality parameters as a function of FIX5 azimuth orientation angle, for the standard FIX5 system flying in upwind direction. Top panel: Results for along-track (u) component. Lower panel: Results for across-track (v) component. Displayed parameters as in Fig. 5.

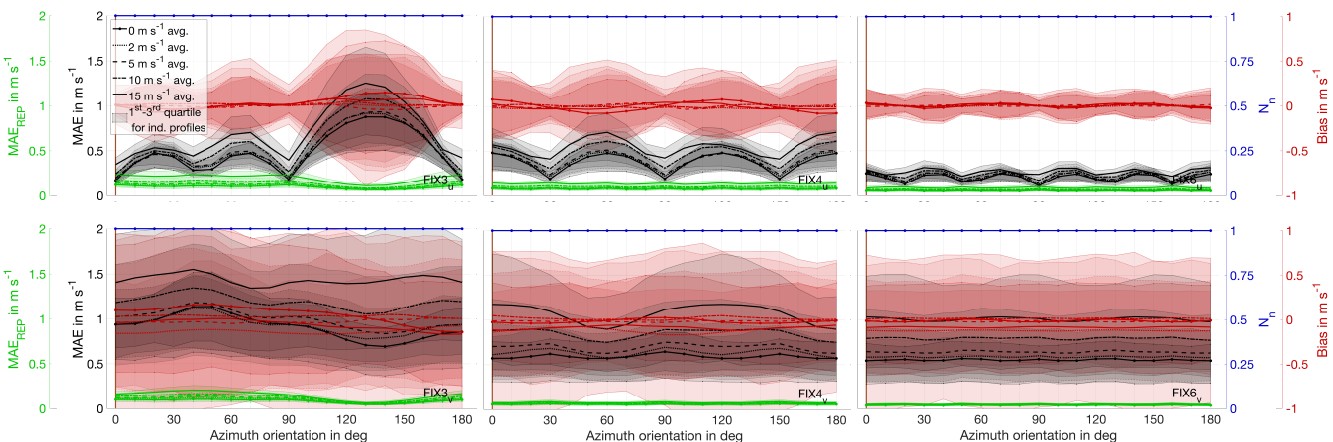

**Figure B9.** Retrieval quality parameters as a function of azimuth system orientation angle for FIX3, FIX4 and FIX6 systems flying in an upwind flight direction. Left panels: FIX3 system. Middle panels: FIX4 system. Right panels: FIX6 system. Top panels: Along-track (u) component. Lower panels: Across-track (v) component. Displayed parameters as in Fig. 5.





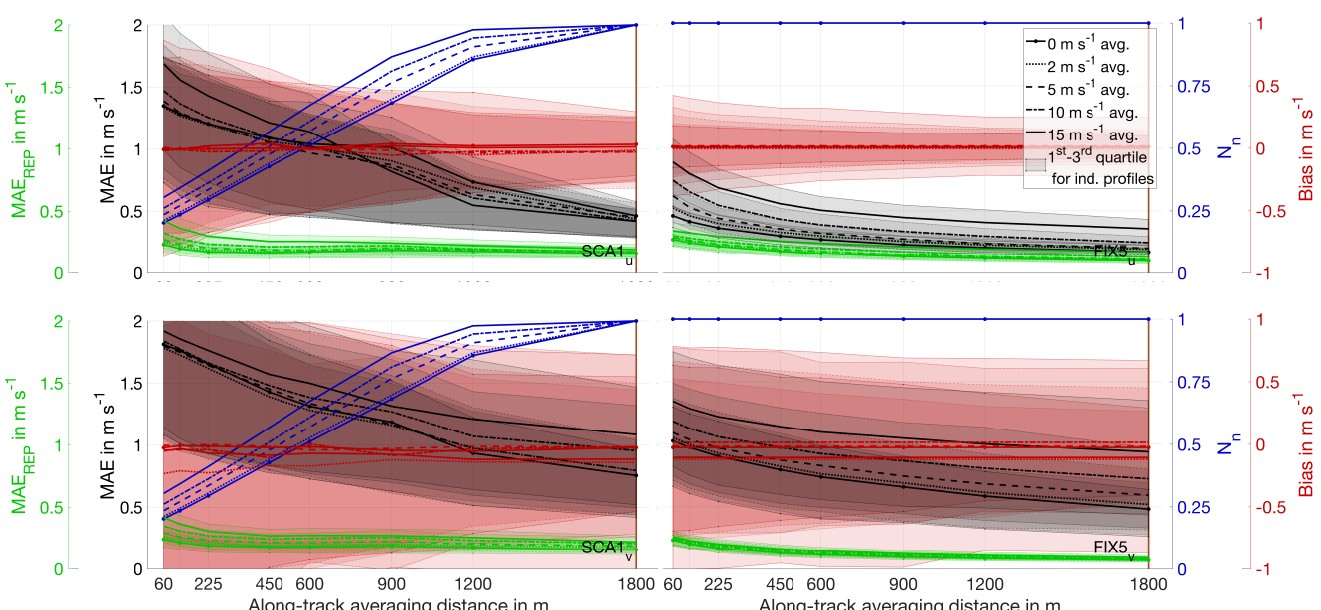

**Figure B10.** Retrieval quality parameters as a function of along-track averaging distance for the standard SCA1 and FIX5 system flying in an upwind direction. Left panels: SCA1 system. Right panels: FIX5 system. Top panels: Along-track (u) component. Lower panels: Across-track (v) component. Displayed parameters as in Fig. 5.



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
