# Peer review of "Advancing airborne Doppler lidar wind profiling in turbulent boundary layer flow - an LES-based optimization of traditional scanning-beam versus novel fixed-beam measurement systems"

_EGUsphere, 2023_

## Referee Comment (RC1)

Review of the article

**"Advancing airborne Doppler lidar wind profiling in turbulent boundary layer flow - an LES-based optimization of traditional scanning-beam versus novel fixed-beam measurement systems "**

submitted by Gasch et al.
**(AMT)**

**Review Summary**

LES simulations are used to compare the performance of airborne Doppler wind lidar measurements with conventional scanning techniques and a multiple fixed-beam approach for probing planetary boundary layer (PBL) winds. The paper is well written and clearly demonstrates a few of the advantages that a fixed-beam system would have especially regarding the reduction of the representativeness error in a turbulent wind field which is particularly interesting as such a fixed-beam approach is easily implementable with novel all-fiber-based laser transmitters. Hence, it is recommended to accept the paper manuscript after addressing the points that are raised below

**General comments**

- The main rationale of the paper manuscript is to compare the wind results retrieved from a fixed-beam approach (1 beam nadir, 4 beams off-nadir) with the one obtained from a scanning approach (continuous scanning with a 20°/s scanner rotation speed). However, a more suitable comparison to a scanning approach would actually be to consider a step-and-stare scanning with 5 LOS measurements (1 nadir and 4 off-nadir) comparable to the one provided by the fixed-beam approach. Considering 1-s for each of the LOS measurements, such a system would provide wind data for all 5 seconds with similar LOS information as available from the fixed-beam measurements, however, with a temporal discrepancy. Similar scanning schemes are already applied to airborne wind lidar systems. Thus, it would be recommended to replace or update the current analysis with such a scanning scheme. This would also further confirm and strengthen the benefit of using a fixed-beam approach for current wind lidar systems.
- Somehow related to this issue: In chapter 4, you investigate the error dependency of the elevation angle and the number of fixed LOS beams. Would a similar optimization procedure also be possible for a (step-and-stare) scanning approach? Probably, also scanning schemes could be optimized for the respective situation and would for instance easily allow correcting for crabbing/crosswind.
- As airborne wind lidars are also often used to probe the entire troposphere, it would be very useful to comment on the performance of a fixed-beam instrument in such a case. Probably there is no need to perform advanced LES simulations for that. But could you at least comment on the wind measurement performance in non-turbulent flow as it is expected in the free troposphere?
- It is appreciated and understood that LES simulations provide a lot of advantages for system optimization. Anyway, wouldn't it be useful to directly compare the performance of a fixed-beam and a scanning system e.g. by ground measurements? Such a comparison would give further insights into the different approaches even without knowing the actual wind field truth. Although it is clear that such kind of measurements does not have to be included in

this paper, it should be stated that such kind of measurements are foreseen to be performed in the future.

**General comments**

- Page 2, line 37: O(10 km): Actually, this is constrained by the "old" scanner motors that are used. With an optimized system, horizontal resolutions of e.g. 4 km are feasible. Furthermore, this option is using 21 LOS! if 5 LOS would be used, the horizontal resolution would be ~1 km. This should be put into context here.
- Page 2, line 54: five independent lidar systems: Probably, you are talking about beams or LOS directions, right?
- Page 2, line 55: What is actually the vertical resolution of the lidar measurements (pulse length) that you are considering? 30 m? Do you use similar assumptions for the fixed-beam and scanning system?
- Page 3, line 83: retrieval error introduced by turbulence: It would be nice to read the number that you get (quantitatively)...how large can this error get, and how large is it typically.
- Page 4, line 98: question → questions
- Page 4, line 108: higher surface sensible heat flux: Is it still a realistic number or is it significantly higher than in the real world?
- Page 8, lines 173 ff: Rotation speed: Is 20°/s the maximum that can be reached without significant losses due to the lag angle? Have you, of curiosity, performed a similar analysis with a rotation speed of e.g. 35°/s? It would be interesting to state if this significantly influences the performance of the retrieved winds.
- Page 8, line 191, "zenith". Shouldn't it be nadir, as you are downward looking?
- Page 8, line 199: along/across-track components: Having these 4 LOS measurements (along and across track) would also be very beneficial for GW research, e.g. flux retrieval as recently shown in Witschas 2023.
- Page 11, line 262: Could you give examples of how the values change for each reference truth? Is it more cm/s or m/s? Would be helpful to read the numbers here.
- Page 11, Eq. 1: What is MAE standing for? Is it mean absolute error?
- Page 23: lines 536 ff: This is only true when equally separating the different LOS. What about an unequal separation e.g. by keeping the fore and aft beams? Have you ever considered such an option?

---

## Author Comment (AC2)

Review comments

Advancing airborne Doppler lidar wind profiling in turbulent boundary layer flow-an LES-based optimization of traditional scanning-beam versus novel fixed-beam measurement systems

Gasch, et al. 2023

Readability: Very well written; well organized and thorough.

Significance: Although the multiple fixed beam (each with its own lidar) is not novel for a proposed and simulated space based DWL (e.g. JEMCDL), the investment in another DWL tool for airborne atmospheric and ocean surface research is highly merited. Some advantages of a 5 FIXED beam (each continuous transmitting and receiving) compared with a scanning ADWL sampling in a cycloidal pattern using just a single transmitter/receiver for a certain subset of observational/research goals are obvious.

Methodology: Use of an LES-based airborne Doppler lidar simulation test bed is ideal for isolating sampling related errors of representing the "true" profile of the wind within a target volume in the presence of wind shear (both speed and directional) and turbulence. The attributions of "error" to turbulence or representativeness is very useful in operating and configuring an ADWL as well as processing the LOS retrievals to obtain estimates of the vertical profile of u and v within a dynamically non-homogeneous target volume, in this case the middle layers of an unstable PBL.

I am recommending that this paper be published after minor revisions. The revision I suggest is to acknowledge that SCA1 concept does not represent ADWL configurations currently in use by NASA, NOAA and ONR. SCA1 represents a simple continuous scanning mode suitable for this initial study.

**We would like to thank Dr. Emmitt for the time and effort taken in reviewing the manuscript.** The discussion of the points raised below certainly improves the quality of our study. We have adapted the manuscript based on the answers given below, also taking into consideration the points raised by the reviewer Dr. Witschas. We hope that the manuscript is acceptable for both reviewers in the revised form.

Exceptions taken: Following is a discussion of exceptions taken to the experimental setup which raises issues with all subsequent conclusions.

1. The evaluation metrics in this paper are wind vector product centric. While this does not invalidate this papers investigation of the merits of FIX5 vs SCA1, a major utility of the ADWL observations is numerical model validation and numerical model Data Assimilation, both of which prefer LOS ADWL retrievals, leaving the full vector wind profile to second tier processing.

If LOS observation density and distribution in terms of along track and cross track directions were used for the basic comparisons, different conclusions could be reached as to which sampling technique serves the modeling community best.

**Answer to 1.:** In our opinion a vast majority of studies retrieve wind profiles from the measured LOS velocities, and we have now included a literature overview of studies doing so. Wind profile retrievals are needed for experimental and process-oriented studies since the pure LOS measurements are near-impossible to interpret with respect to their physical meaning. Hence, we put the focus on wind profile retrieval accuracy. Certainly, numerical model data assimilation could use the measured LOS observations directly, but we are unaware of studies which have done so up to date and therefore put the focus on wind profiling quality. We believe that a FIX5 system also is beneficial for the modelling community for three reasons: First, the system provides five times the amount of LOS data compared to a scanning beam system, allowing for better statistics and a more complete exploration of the retrieval volume. Second, the five stare directions are available without interruption, allowing for more reliable estimation of turbulent fluctuations alongside a simpler forward operator to compare model equivalents. Third, for a fixed-beam system the aircraft motion correction accuracy (largely determining LOS accuracy) does not depend on the pointing accuracy of the scanner and hence is expected to be more reliable. Additionally, we hope that the simpler fixed-beam design will allow more widespread and cost-effective ADL measurements, which will strengthen model evaluation and data assimilation in general.

**Changes in the manuscript:**

- Included literature overview on wind profiling studies.
- Since our study is focused on wind profiling retrieval accuracy we would like to avoid including statements or a discussion on model studies and data assimilation although we believe that the FIX5 system also offers advantages for these communities.

1. The scanning system (SCA1) does not represent the standard (traditional?) sampling pattern used with the ADWLs used in the studies referenced (Bucci, De Wekker) nor the ADWL used on the NASA DC8 (Turk, Kavaya). The coherent ADWLs being used in the USA in large field campaigns over that last 2 decades use two types of scanning:

    1. Fixed elevation with azimuthal scanning in a step-stare mode using 2 -13 azimuthal programmable stops (NASA's DAWN and the new AWP which includes a nadir staring option)

    2. ONRs (also used by NOAA) cylindrical side mounted scanner allowing programmable beam pointing routines within a large azimuth/elevation bounded target volume.

2. The use of a continuous scanning approach (e.g. at 20 degrees/sec) has been replaced with a "step and stare" strategy for many years and for several reasons (lag angle and the desire to eliminate the angular spread in lidar shots being integrated before preforming a spectral analysis).

3. A better (and more relevant) "reference" SCA1 for this sampling centric study would be the following based upon more than 1000 flight hours of observations using the cylindrical scanner:

    1. Elevation angle from the horizontal: 60 degrees

    2. 12 azimuthal stares for 1 second with 30 degree azimuth increments

    3. Slew rate between stares (30 degrees/second)

4. 1 nadir stare (five-10 seconds) in the middle of the 12 stare VAD.

5. 50m range resolution

6. 50 -100m along track averaging.

**Answer to 1., 2., 3.:** Summarizing the above 3 points, we agree that the continuous scan pattern based on G20 was idealized. We did not simulate a step-and-stare approach previously since the lidar internal signal processing is not simulated due to the idealized instrument simulation. Hence, staring does not improve radial velocity signal quality, as it would in a real-world system. Although continuous scans have been used (Augere et al., 2017) we acknowledge the point that the simulated scan pattern should correspond to more often used settings.

Based on the above points and the literature overview conducted by us we have replaced the former SCA1 pattern with a 13 point step-and-stare pattern (SNS13), with scan settings as suggested above. The new settings are as follows:

1. Elevation angle from the horizontal 60 degree.
2. 12 azimuthal stares for 1 second with 30 degree azimuth increments.
3. Negligible slew time between subsequent stares, further improving the step-and-stare pattern wind profile retrieval quality due to faster turnaround times.
4. 1 nadir stare of 1 second in addition to the 12 azimuthal stares (hence SNS13). The nadir time was shortened to 1 s to allow wind profile retrieval with reasonable turnaround times.
5. 30 m range resolution as before (higher than suggested, based on the expected laser performance of the fixed-beam system).
6. 100 m averaging for the radial velocity measurements for each stare direction (1 s stare time at 100 m/s). Averaging does not influence the radial velocity measurement accuracy since the lidar internal signal processing is not simulated (idealized instrument). The along-track averaging of the wind profile retrieval volume is varied between 60 m and 1800 m as before.

**Changes in the manuscript:**

- SNS13 scan pattern used and discussed including literature overview.

4. Had the SCA1 sampling pattern described in 3. above been used for quantifying the advantages (and disadvantages) of the FIX5 system, the following conclusions and expectations might be reversed or at least quantitatively changed.

**Answer to 4.:** We now simulate the scan pattern suggested above and thereby feel confident to address the below questions.

1. Line 35: The Goodness of Fit (GOF) value for a 12 look step stare solution for u,v provides a very useful measure of the non-uniform distribution of winds in the retrieval volume. This GOF is used to generate a confidence metric for representativeness. By performing triple pass processing a reasonable description of the non-uniformity can be made…not assumed except for the first pass.

**Answer to 4.1:** We assume that the GOF refers to the coefficient of determination ($R^2$) parameter which can be obtained from the inversion based fit to the measured radial velocities. However, as shown by G20, using the $R^2$ parameter for quality filtering can introduce unwanted bias in the wind speed retrieval at low wind speeds. Introduction of this bias by quality filtering occurs due to the mapping of vertical wind inhomogeneities into horizontal wind, it is explained in detail in G20. The same behavior described there is evident in the present study. Even without filtering with the $R^2$ the wind speed retrieval is biased at low wind speeds. Introducing $R^2$ filtering severely increase the bias but does not help in bringing the MAE down. To avoid a strongly biased wind speed retrieval, we avoid using the $R^2$ as a quality filtering parameter in this study.

**Changes in the manuscript:**

- Included statement in retrieval section.

2. Line 54: It is not clear why the simulation was not performed for an aircraft flying 500-1000 meters above the PBL top since that may be the preferred perspective on the PBL. For the reasons stated elsewhere in the paper, the "saftest" portion of the PBL to use for analysis is the 100-1000meter layer (middle of the PBL). That is understandable, but the horizontal data coverage from 3000m will be different than from 1500m.

**Answer to 4.2:** The aircraft is flying at 1500 m, the PBL height is 1100-1400 m. While the aircraft is flying at 1500 m, only wind profiles from 100-1000 m altitude are considered for the analysis, e.g. with the distance of 500 m as suggested. The reason is that we want to investigate the impact of turbulence on wind profiling error. Above 1000 m turbulence starts to decay noticeable in the LES since the PBL entrainment and detrainment zone is reached. As the higher altitudes do not represent turbulent conditions, we exclude them in the retrieval. Of course, higher flight altitudes can be simulated in the ADLS, they do not change the results significantly. For example, a higher flight altitudes of 1800 m leads to a slight reduction (< 10%) of error, whereas a lower flight altitude of 1100 m leads to a slight increase (<10%) of error. Flight level changes affect both scanning and fixed-beam approaches equally, hence changing flight altitude does not change the findings from our study.

3. Line 94: Are there any disadvantages of the FIX5 vs the SCA1 for PBL research? Would any of the stated advantages of the FIX5 ADWL change if the more relevant scanning ADWL configuration were used?

**Answer to 4.3:** We have not discovered relevant disadvantages of the FIX5 system so far. One thing that requires attention in real-world measurements is the inability to compensate the aircraft pitch, roll and yaw movement when using fixed-beam directions. However, this is deemed unproblematic as it can be corrected in post-processing and is standard for airborne Doppler radar measurements (Strauss et al., 2015; Gasch, 2021). In addition, an active stabilization of the nadir telescope may also be possible. Also, compared to a scanning system azimuthal resolution of the radial velocity measurements is lost. However, since the azimuthal radial velocity measurements are usually strongly correlated not much information is lost and we do

not see a disadvantage in this. Additionally, the FIX5 system provides five times the amount of radial velocity information in general, since the five beams measure simultaneously.

4. Line 175: Step and Stare scanning greatly reduces the lag angle losses. This is not an issue for the reported study, though.

**Answer to 4.4:** We agree. We now simulate a SNS scan pattern, but since the simulation does not include the backscattering process lag angle losses were not an issue beforehand also.

5. Lines 330-365: Throughout this paper there are frequent references to the issue of alignment of the FIX5 scanning telescopes with the aircraft ground track (crabbing) and the wind direction. With the 12 look scanning ADWL (let's call it SCA2), there are numerous subsets of azimuth look angles that can be used for sector wind vector retrievals, for example, quadrant retrievals. Based upon the 4 wind profiles thus obtained, horizontal gradients and other estimates of non-uniform flow can be deduced and quantified. The presence of PBL jets and directional shear layers does not impact (degrade) the accuracy of the SCA2 profile retrievals as much as might be the case for the FIX5. This point raises issues with all subsequent conclusions.

**Answer to 4.5:** We now simulate a SNS scan pattern as suggested. Certainly, looking at the suggested quadrant retrievals may be interesting for future studies. However, based on the current results it is unclear if reliable retrieval of horizontal gradients is possible. To allow inference of gradients beyond the uncertainty of the retrieval error, the gradients probably would have to be very large. In the results seen so far, decreasing the retrieval volume size results in fewer measurements per retrieval volume and less azimuthal spread, causing a strong increase in retrieval error. Unfortunately, PBL flow is heterogeneous down to very small scales, thereby the homogeneity assumption used in the retrieval is not necessarily fulfilled better. Based on the current results we expect a similar behavior for the quadrant retrieval approach.

**Changes in the manuscript:**

- We now state that optimizing step-and-stare approaches as well as modified retrieval strategies also offers interesting potential and is worth investigating in the future (scanning section and conclusions).

6. Lines N/A: This paper does not discuss explicitly the ability to measure vertical velocity. The primary interest is in the impact that very local vertical motions associated with organized structures such as OLEs will have on the calculation of horizontal wind components. However, the DC offset of the 12 point solution along with the 5 – 10 second vertical stare provide insight to the scales of vertical "contamination" of the horizontal wind retrieval. This, at the least, provides a means of attaching "quality" flags to each profile. Without this "over sampling" compared to only 2-4 perspectives, there is much less basis for judging representativeness in actual applications.

**Answer to 4.6:** We understand that this point suggests using the residuals from the radial velocity based wind profile retrieval for an uncertainty analysis, in addition to the vertical wind measurements. We have tried to do so in Gasch (2021) and this is certainly an excellent point for future study. So far, it is quite complicated to convert the scales of contamination into a quantitative uncertainty estimation. In any case, data-driven uncertainty estimation should be possible with higher accuracy for a FIX5 system compared to a scanning system. More radial velocity information is available from a FIX5 system, hence the scales can be determined more reliably. We have decided against including a 5-10 second vertical stare in the step-and-stare pattern used in this study, since it would have increased scanner turnaround times and thereby coarsened the along-track resolution of the wind profiles noticeably. Instead, we include a 1 s vertical stare to match the duration of the other look directions.

**Changes in the manuscript:**

- We believe that an evaluation of the possibility of a data-driven uncertainty estimation is beyond the scope of this study but it has been mentioned by G20 as worthwhile to explore already.

1. In spite of the exceptions and concerns expressed in 5. above, this paper is well written and answers the questions (Lines 94-97) based upon the assumed SCA1 configuration. However, to be relevant to how ADWLs have been scanned for that last 20 years (like ONR's TODWL and NRL/NOAAs P3DWL with very flexible pointing options) and are being designed for the next generation of high energy ADWL (like NASA's AWP with a vertical stare option), there is the need to simulate a SCA2 type instrument and ask the questions a slightly different way. 1.) How do the FIX5 and SCA2 serve the modeling community vs the atmospheric processes community? (i.e. LOS as primary product vs wind vector profile?) 2.) Do fewer but more accurate bi and quad perspective profiles over smaller foot prints trump more LOS samples from more (say, 12) perspectives, especially in complex flows?

**Answer to 1.:** We have implemented the modified SNS scanning system as suggested and have addressed the points raised above. We also address the discussion on the value of wind profile vs. radial velocity measurements above. The additional points mentioned certainly provide excellent points of investigation for future studies but are beyond the scope of this study and would require more than minor changes, as suggested.

Regardless of any follow-on simulations, the development of the FIX5 instrument has merit and will not only provide collocated (for no crabbing) bi-perspective simultaneous LOS measures of the winds, but will also use SOTA fiber laser technology for each telescope and, hopefully, a less expensive and thus more available means of making airborne wind profiles for both academic research as well as applications. The increased number of vector profiles per km along a track is certainly attractive.

**We would like to thank Dr. Emmitt again for taking the time and making a great effort in reviewing this manuscript. We are certainly excited to see the first measurements from the fixed-beam system and we hope that it will serve the community by providing new opportunities and insight.**

---

## Author Comment (AC3)

Review of the article

**"Advancing airborne Doppler lidar wind profiling in turbulent boundary layer flow - an LES-based optimization of traditional scanning-beam versus novel fixed-beam measurement systems "**

**Review Summary**

submitted by Gasch et al.

**(AMT)**

LES simulations are used to compare the performance of airborne Doppler wind lidar measurements with conventional scanning techniques and a multiple fixed-beam approach for probing planetary boundary layer (PBL) winds. The paper is well written and clearly demonstrates a few of the advantages that a fixed-beam system would have especially regarding the reduction of the representativeness error in a turbulent wind field which is particularly interesting as such a fixed- beam approach is easily implementable with novel all-fiber-based laser transmitters. Hence, it is recommended to accept the paper manuscript after addressing the points that are raised below.

**We would like to thank Dr. Witschas for the time and effort taken in reviewing the manuscript.** The discussion of the points raised below certainly improves the quality of our study. We have adapted the manuscript based on the answers given below, also taking into consideration the points raised by the reviewer Dr. Emmitt. We hope that the manuscript is acceptable for both reviewers in the revised form.

**General comments**

- The main rationale of the paper manuscript is to compare the wind results retrieved from a fixed-beam approach (1 beam nadir, 4 beams off-nadir) with the one obtained from a scanning approach (continuous scanning with a 20°/s scanner rotation speed). However, a more suitable comparison to a scanning approach would actually be to consider a step-and- stare scanning with 5 LOS measurements (1 nadir and 4 off-nadir) comparable to the one provided by the fixed-beam approach. Considering 1-s for each of the LOS measurements, such a system would provide wind data for all 5 seconds with similar LOS information as available from the fixed-beam measurements, however, with a temporal discrepancy. Similar scanning schemes are already applied to airborne wind lidar systems. Thus, it would be recommended to replace or update the current analysis with such a scanning scheme. This would also further confirm and strengthen the benefit of using a fixed-beam approach for current wind lidar systems.

**Answer:** Our study aims to quantify the advantage of the novel-fixed beam design compared to **traditional ADL systems** available **for PBL research** up to date, as stated in the title of the study. For this reason, we used the scanning system with comparable settings as those used by Gasch et al., 2020.

To provide more context on used scan pattern to the reader, we have now included a literature overview on reported scan techniques and an extended discussion. In addition, based on this review and taking into consideration the comments made by the reviewer Dr. Emmitt, we have now switched from the continuous scan pattern to a step-and-stare (SNS) pattern with a much faster scanner. The new SNS pattern is based on scan pattern which have been reported in PBL literature before (also see answers in Dr. Emmitt's review). To enable a better comparison, we have added an additional vertical

stare to the twelve-point SNS pattern as suggested in the above comment. Thus, we now use an overall thirteen-point SNS pattern (SNS13) with a rapid scanner as a reference system.

We are unaware of a system with a very rapid 5 s SNS scan (SNS5), as proposed in your comment, having been used for wind profiling on slow and low flying aircraft for PBL research up to date. Since it hasn't been used in PBL research up to date, we consider the fast SNS technique mentioned by Dr. Witschas to be another promising approach, which may be possible to build in the future. One possible solution appears the use of double-wedge scanners (as used by Dr. Witschas on fast jet aircraft), although double-wedge scanners may be limited in their ability to scan at shallow elevation angles (an important driver of retrieval error).

For your reference, we have also simulated the SNS5 pattern and attach the results to this reply. While wind profiling quality and availability is slightly better than for the SNS13 pattern, the retrieval availability and accuracy still show reduced quality compared to the FIX5 system. Especially at short averaging distances < 450 m the profile availability is strongly reduced. The reduction in error levels for some background wind speeds at very short averaging distances is due to the $CN_{max} = 10$ quality filtering, which leaves very few but well sampled retrieval volumes. In addition to the retrieval resolution loss, an increase in error levels is observed for both wind components if steeper scan elevation angles are used. Additionally, quality filtering with $CN_{max} = 10$ results in a loss of wind profile points at the steepest elevation (80 degree) compared to the SNS13 pattern. Overall, having the highest resolution and lowest error possible is crucial for PBL studies in order to resolve the spectrum of turbulence down to the inertial subrange. Hence, we believe that the improved wind profiling capabilities of the novel fixed-beam approach are needed.

Overall, our study does not claim to be exhaustive and investigate all possible improvement options. We believe the fixed-beam technology provides unique advantages and is an important step forward, which is why we are building and investigating such systems. There is also potential merit of the very fast SNS pattern, but we believe that this merit needs to be carefully examined in a future study. Some of the pros and cons go beyond the LES-based wind profiling accuracy analysis and require in-depth discussion, which is beyond the scope of this study and would extend beyond the minor changes suggested:

- A fixed-beam system enables a simplification of the rack design as no moving parts are needed. This simple design is beneficial for system design, certification and reliability and may enable easier transfer of the Doppler lidar technology between different carrier aircraft.
- No measurement time is lost due to slewing for fixed-beam systems.
- It is important to have uninterrupted vertical wind measurements inside the PBL at high resolution to include and resolve the inertial subrange in the measurements. Uninterrupted vertical wind measurements, yielding highest spatial resolution, are only available from a fixed-beam system.
- Since the inertial subrange is resolved it may be possible to retrieve turbulence using a fixed-beam system. We investigate the possibility to retrieve turbulence using the fixed-beam system in an upcoming study (Kasic et al., in preparation).
- For a SNS system, the additional design, manufacturing and certification of a scanner unit is necessary, which is expensive. As an advantage, only a single lidar unit is needed in a SNS system, which can therefore potentially use more expensive technology enabling higher lidar radial velocity measurement quality. However, the lidar beam is attenuated by the scanner, reducing the benefit of the more expensive lidar unit. The fiber-based lidar units to be used in the fixed-beam system offer sufficient measurement quality for PBL turbulence measurements where sufficient return signal is available. Hence, the need for using more expensive lidar units is marginal. Additionally, the fixed-beam directions offer the advantage of situation (aerosol return) dependent averaging times.

- A multiple fixed-beam system can still measure if one of the individual lidar units fail, whereas a single lidar and single scanner present single points of failure.
- A SNS system may allow for correction of the aircraft crabbing angle for improved curtain retrievals. However, correcting the crabbing angle will not solve issues caused by wind advection between measurements conducted at different times on the curtain below the aircraft (see our answer to the next question).
- A very fast scanner makes careful investigation of the pointing angle accuracy necessary. This issue is not considered in our current idealized simulation since beam pointing angle accuracy is stable for rigid fixed-beam systems.
- To our knowledge, double-wedge scanners offer limited possibilities for shallow scan angles, which is an important parameter for wind profiling quality, as shown by our results.

Based on the reasoning detailed above we have thus decided to compare the FIX5 approach with the traditional SNS13 approach in this study.

**Changes in the manuscript:**

Based on the above explanations we have adapted the manuscript to acknowledge the point and perspective presented by both reviewers:

- Based on the literature overview and comments by Dr. Emmitt we have adapted the reference scan pattern to the fast SNS13 commonly used for wind profiling in the PBL before.
- We now state that an even faster SNS approach also offers interesting potential and is worth investigating in the future (introduction, scanning section and conclusions).

- Somehow related to this issue: In chapter 4, you investigate the error dependency of the elevation angle and the number of fixed LOS beams. Would a similar optimization procedure also be possible for a (step-and-stare) scanning approach? Probably, also scanning schemes could be optimized for the respective situation and would for instance easily allow correcting for crabbing/crosswind.

**Answer:** Yes, the system setup optimization and error characterization are also possible for a SNS approach. Having switched to the SNS pattern in the manuscript we do so now for the beam elevation angle. While using a SNS approach would allow to correct for the aircraft crabbing angle (e.g. better aligning the forward and aft measurements on the ground track below the aircraft), it would still not be able to avoid effects caused by advection due to the time elapsed between the measurements. Advection effects depend on the speed and direction of the wind, which is also a function of altitude, in relation to the flight direction. The combination of both is complicated to consider, as they can both act in favor or against better alignment of the measurements. Already relatively small advection distances O(100 m) can cause noticeable retrieval error, since the integral scale of turbulence in the PBL is small. Additionally, as mentioned in the discussion above, pointing accuracy issues would have to be considered in more details for a fast SNS pattern.

**Changes in the manuscript:**

- We now mention the scanning-beam optimization as an interesting future point of study.

- As airborne wind lidars are also often used to probe the entire troposphere, it would be very useful to comment on the performance of a fixed-beam instrument in such a case. Probably there is no need to perform advanced LES simulations for that. But could you at least comment on the wind measurement performance in non-turbulent flow as it is expected in the free troposphere?

**Answer:** We expect that a fixed-beam system also offers advantages above the PBL. While retrieval error due to turbulence become less important above the PBL (but can still be considerable, see Weissmann et al., 2005), signal availability becomes increasingly important. In the free troposphere less scatterers are available, decreasing the signal to noise ratio. For this reason, longer accumulation times are needed to obtain useful radial velocities (more spectral averaging), which is also a main motivation for using SNS scanning approaches instead of a continuously scanning beam. This explains the expected benefit of the fixed-beam approach: Since all radial velocities are available continuously and with a stable aircraft speed projection, the needed averaging times to obtain a useful signal can be chosen individually for each beam, depending on the aerosol return available. In addition, no measurement time is lost due to the slewing of the scanner. Last, the continuous availability of the nadir beam for uninterrupted high resolution vertical wind measurements is expected to be beneficial for many application scenarios.

The LES-based simulator presented here does not include a simulation of the backscattering process and spectral analysis, therefore it cannot be used to assess the benefits of a fixed-beam approach on signal quality above the PBL. We hope to be able to provide more answers once the first measurements with the real system have been conducted, since the results depend on the signal quality of the fiber-based lidars.

**Changes in the manuscript:**

- We state that the simulation does not include a simulation of the backscattering process, hence we would like to not comment on free troposphere behavior as it is beyond the scope of this study.

• It is appreciated and understood that LES simulations provide a lot of advantages for system optimization. Anyway, wouldn't it be useful to directly compare the performance of a fixed-beam and a scanning system e.g. by ground measurements? Such a comparison would give further insights into the different approaches even without knowing the actual wind field truth. Although it is clear that such kind of measurements does not have to be included in this paper, it should be stated that such kind of measurements are foreseen to be performed in the future.

**Answer:** Real-world validation is certainly key besides the LES-based simulation. We do not see these two aspects as mutually exclusive but as inclusive instead. The LES-based simulations allow us to study system setup and retrieval strategy optimization before building and flying a real-world system. Since system design, manufacturing and flight test are expensive, this is desirable. Of course, any real-world system also needs to be validated with other measurement systems (e.g. ground-based, dropsondes, other aircraft), since they can suffer from a variety of other error sources not considered in the idealized LES-based simulation. An important example is the motion correction error (linked to beam pointing-angle calibration). Fortunately, real-world measurements also offer further validation diagnostics such as the lidar measured ground-return velocity. For this reason, extensive real-world validation of ADL systems have been performed in the past, and this will also be key for the fixed-beam systems under development. However, we believe flying and validating a whole range of elevation angles with a sufficient statistical basis is unfeasible due to the prohibitive cost. Hence, the ADLS optimization has been conducted.

**Changes in the manuscript:**

- Included statement on the need of real-world validation in conclusions.

**General comments**

- Page 2, line 37: O(10 km): Actually, this is constrained by the "old" scanner motors that are used. With an optimized system, horizontal resolutions of e.g. 4 km are feasible. Furthermore, this option is using 21 LOS! if 5 LOS would be used, the horizontal resolution would be ~1 km. This should be put into context here.

  We deemed this a rather recent study, therefore included it in this way. Reworded and added a very recent reference now, although only partial wind components are retrieved by the faster scan pattern and it has not been applied to PBL wind profiling so far. Also included a literature review now, see above comments.

- Page 2, line 54: five independent lidar systems: Probably, you are talking about beams or LOS directions, right?

  Yes, clarified now.

- Page 2, line 55: What is actually the vertical resolution of the lidar measurements (pulse length) that you are considering? 30 m? Do you use similar assumptions for the fixed-beam and scanning system?

  Yes, the vertical resolution is 30 m, corresponding to an expected (selectable) pulse length of 160 ns for the fixed-beam lidar systems. The 30 m resolution is equal for the fixed-beam and scanning system in our study. Since this is a rather technical information we would like to not discuss it in the introduction, we have now added this information in Tab. 2.

- Page 3, line 83: retrieval error introduced by turbulence: It would be nice to read the number that you get (quantitatively)...how large can this error get, and how large is it typically.

  The retrieval error very much depends on scan settings used and turbulent conditions present. We added a statement now to specify the order of magnitude.

- Page 4, line 98: question→questions

  Changed, thank you.

- Page 4, line 108: higher surface sensible heat flux: Is it still a realistic number or is it significantly higher than in the real world?

  This is a rather common surface sensible heat flux for a daytime continental convective PBL, now mentioned.

- Page 8, lines 173 ff: Rotation speed: Is 20°/s the maximum that can be reached without significant losses due to the lag angle? Have you, of curiosity, performed a similar analysis with a rotation speed of e.g. 35°/s? It would be interesting to state if this significantly influences the performance of the retrieved winds.

  It is our experience from real-world measurements that the SNR starts to degrade noticeably beyond 20°/s scan speed and becomes not usable beyond 45°/s. However, since we have not investigated this theoretically or systematically, we would like to avoid making statements in this direction. We now state that scan pattern optimization is an interesting point for future study.

- Page 8, line 191, "zenith". Shouldn't it be nadir, as you are downward looking?

  Absolutely, changed.

- Page 8, line 199: along/across-track components: Having these 4 LOS measurements (along and across track) would also be very beneficial for GW research, e.g. flux retrieval as recently shown in Witschas 2023.

  We have added a reference stating this.

- Page 11, line 262: Could you give examples of how the values change for each reference truth? Is it more cm/s or m/s? Would be helpful to read the numbers here.

  It is O(cm/s) but depends on altitude as well as scan pattern and retrieval strategy investigated. Generally speaking differences between triangular and square volume truth are always small with O(0.05 m/s). They are largest towards the top of the boundary layer where the truth volumes differ due to the triangular versus square shape of the volume.

  As expected, the nadir truth can show reduced error for the fixed-beam setup when the retrieval quality of the along-track components is investigated (O (0.2 m/s)). In this case, the measurements are conducted in a very confined volume close to the nadir truth. Thus, the retrieval error is lowered for the along-track components, but increased for the across-track components, when using the nadir truth. Since the nadir truth favors one component over the other we have decided on using the triangular volume reference truth which incorporates both retrieved components equally.

  We've included a statement on the order of magnitude in the revised manuscript now.

- Page 11, Eq. 1: What is MAE standing for? Is it mean absolute error?

  Yes, it is the mean absolute error, now stated also.

- Page 23: lines 536 ff: This is only true when equally separating the different LOS. What about an unequal separation e.g. by keeping the fore and aft beams? Have you ever considered such an option?

  So far we only simulated fixed-beam system with equidistant spacing for two reasons. First, the equidistant spacing maximizes the information added by each beam. Orienting beams closer together will lead to increased correlation between the measurements from different beams. Second, not constraining the system setup variation to symmetric options theoretically opens an infinite number of options due to the flexibility in azimuth and elevation. We believe this would be quite indigestible by the reader and therefore try to provide generalized results from which the effect of specific setups can be inferred. Our main lessons to do so are:

  - Shallow elevation measurements and spatially co-located measurements help reduce retrieval error.
  - Increasing the number of beams helps reduce azimuthal variation of retrieval characteristics and lowers the retrieval error overall.

**We would like to thank Dr. Witschas again for taking the time and making a great effort in reviewing this manuscript. The comments provided above have certainly improved the quality of our manuscript and the discussion.**

**1 SNS5 vs. SNS13**

[Figure]

Figure 1: Retrieval quality parameters as a function of beam elevation angle for the standard SNS13 and SNS5 system flying in crosswind direction. Left panels: SNS13 system. Right panels: SNS5 system. Top panels: Across-track (u) component. Lower panels: Along-track (v) component.

[Figure]

Figure 2: Retrieval quality parameters as a function of along-track averaging distance for the standard SNS13 and SNS5 system flying in crosswind direction. Left panels: SNS13 system. Right panels: SNS5 system. Top panels: Across-track (u) component. Lower panels: Along-track (v) component.

---

## Author Response (AR2)

**Review comments by Dr. Witschas**

The revised paper manuscript by Gasch et al. impressively demonstrates how respective scan patterns (or fixed beam configurations) impact the accuracy of retrieved wind speeds from airborne wind lidar instruments. The research is well presented, and hence, it is recommended to publish the paper after minor technical corrections.

In section scanning-beam setup it is stated that Witschas et al., 2023 using three LOS directions, however, only two are used, namely the fore and back propagating beams with an off-nadir angle of 20°. This should be corrected.

Furthermore, in Table A1, Schäfler et al. is cited for a scan pattern different from the VAD scan and the nadir pointing. I am not aware of which scan pattern was applied. It would be useful to cite Witschas et al. 2023 here, as the fore and aft scan was applied to the mentioned 2-µm DWL.

Apart from that, I have no further comments and I am looking forward to seeing the paper published in AMT.

Answer:
With we would like to thank Dr. Witschas for the detailed reading of the revised manuscript and the helpful corrections, which we have included in the manuscript.

**Comments from a second review of:**
Advancing airborne Doppler lidar wind profiling in turbulent boundary layer flow-an LES-based optimization of traditional scanning-beam versus novel fixed-beam measurement systems Gasch, et al. 2023

We would like to thank the reviewer for the remarks on the revised manuscript and the detailed information. Addressing the points raised below has certainly improved the quality of our manuscript.

The authors responded to my comments and have made a better case for a FIX5 ADWL being added to our airborne options for remote sensing of the atmospheric PBL. As with any reading of a manuscript multiple times there still remain a few questions. Since this paper is, in large part, lidar technology neutral and is primarily a sampling sensitivity study, the issues of PRF, EAP and backscatter weighting in sample integration required (or desired) are set aside (Line 235). In practice, it may be neither desirable nor necessary to fly within the PBL or even just above the PBL. The trapezoidal "truth volume" is defined by the flight altitude and scanning geometry. In practice, a much more capable (EAP) lidar is required to deal with the R*R losses by flying higher and thus requiring significant hardware and optical resources to have multiple perspectives illuminated with individual lasers. This reality is further in force when considering going to space. I only mention this since I do not think these simulation results should be mistaken for a general conclusion regarding scanning a high EAP lidar vs using multiple lower EAP lidars in a fixed configuration.

Answer: We agree that real-world measurements are subject to a number of additional challenges not addressed in our simulation study which we mention repeatedly. The focus of the study on airborne wind profiling in the turbulent boundary layer is stated in the title, therefore we think that confusion with other Doppler lidar applications is unlikely. We do not suggest direct transfer of the results to other applications (e.g. ADL systems on fast jet aircraft above the PBL or in space) in our study.

In order to provide more clarification and context to the reader we have now included a section addressing the scope of the lidar simulation in the lidar simulation section (see also our answers below). Further, we have extended the list of points relevant for ADL performance in real-world measurements in the conclusions.

As a side note, we think that flying within the boundary layer or just above is frequently applied and beneficial for boundary layer studies. With such a flight pattern below cloud base sampling using high-resolution in-situ measurements can be achieved, which has been conducted frequently in the past.

That aside, I have 2 points that the authors may want to address in their final submission:
1. What was the assumed PRF for the SNS13 scan type? 10Hz as appears to be the case for the FIX5 in Table 2? If so, then the spacing between samples in the along track direction during each of the 12 stares would be ~10m flying at 100m/s. Right?

Answer to 1: Good point, for the lidar simulation we referred to G20 and hence did not make this sufficiently clear. In line with G20, we do not specify a PRF in our study since the lidar signal processing is not simulated. Hence, we only specify a frequency at which radial wind

data is available. We now give a more extensive explanation in the 'Lidar simulation' section and renamed this frequency to 'data rate' in order to avoid confusion with the PRF (see also our answer above). The data rate is 10 Hz for both the SNS13 and the FIX5 system. A 10 Hz data rate corresponds to 10 m sampling distance for each individual radial velocity measurement, since the aircraft is flying at 100 m/s. For the SNS13 pattern the stare duration for each stare direction is 1 s, e.g. averaging 10 radial velocity measurements per viewing direction (see the revised Fig. 2).

   2. The quality metric discussion (lines 289 – 305) still contains several possibly confusing concepts in spite of the authors' efforts to explain the difference between MAE, MAErep and MAEturb.

Answer to 2: Thank you for bringing our attention to this, we now clarify that the calculations are conducted exemplarily for the u component but equal for the wind components. Further we now specify the meaning of the index i, which refers to the individual wind profile points.

a. As I understand it, it takes the SNS13 13 seconds to complete a scan with 1 second dwells that will then be processed into a vector wind profile with 30m vertical resolution. The difference between the two computed horizontal wind components (w assumed zero and thus an error source) at each vertical level and the trapezoidal "beam truth" at those same levels contributes to the MAE expression (Eq 1). "N" is the number of complete profiles achieved while "flying" the 8 transects shown in Figure 1 which would yield N ~ 144.

Answer to 2a: We now clarify the meaning of the formulas and explain the used variable in more detail. We also have corrected N to read $N^R$. $N^R$ refers to the overall number of wind profile points available for each system setup and retrieval strategy, e.g. 67200 for the standard case (see Fig. 3). As written above, we now clarify that the MAE calculation is exemplarily conducted for the u component but valid for the other wind components as well. We hope that this avoids the confusion which occurred previously.

It is important to note that we do not assume zero vertical wind in the retrieval but instead retrieve w as well (see Fig. B3 where the retrieval quality for the w component is shown). We did not explain the retrieval procedure clearly enough in the retrieval section and did not include the reference to G20. We have now corrected this omission in the 'Retrieval strategy' section and Table 2.

b. It appears that the 1 second dwell used by SNS13 in practice is replaced (Line 245) with distance integrations between 60 and 1800m. Is this integration per LOS perspective? Or is it the distance the plane flies before generating another vector wind profile? If so, then the number and location of SNS13 radials used to represent the LES domain samples will vary while only the along track length of the FIX5 integration lines will vary.
c. I can see how the FIX5 system can be programmed to generate a complete profile every so many meters since all 5 lidars are in constant operation.

Answer to 2b and 2c: Based on the changes to your point 2a have now included the correct variable names in Sec. 5 to provide more clarity for the reader.
The retrieval procedure is equivalent for both the SNS13 and the FIX5 system. The along-track sampling distance specifies the distance over which radial velocity measurements are

considered. It is defined as a ground-relative volume (see Fig. 1 and Fig. 2), e.g. all lidar stares that fall into a retrieval volume are considered, irrespective of the aircraft position from which they were conducted. For example, measurements by forward and backward stares occur before and after the aircraft passes over a given ground-based retrieval volume, respectively. As stated by the reviewer, for shorter averaging distances less radial velocity measurements fall into the retrieval volume both for the SNS13 and the FIX5 system (both system always provide 10 Hz data rate in the simulation). For the SNS13 system at short averaging distances, a sufficient number of radial velocity measurements from different azimuth positions is rarely available in the retrieval volumes. Hence, the number of retrievable profiles $N^R$ is severely degraded compared to $N^T$ (CN filtering removes unreliable wind profile values, e.g. when the azimuthal spread of the lidar measurements is too small, leading to a collinear retrieval matrix, see Appendix B). To avoid the issue of non-retrievable retrieval volumes the along-track averaging distance is usually set to correspond to the distance covered by the aircraft during one scan revolution (thus covering all azimuthal positions) in real-world measurements.

As you write, the FIX5 system allows wind profile retrieval even at short along-track averaging distances, since all retrieval volumes contain measurements from 5 different azimuth positions.

Based on our answers and corrections provided above we hope that the retrieval strategy and quality metrics calculation are clear now.

d. With the frozen turbulence assumption, how is it that a system that makes 12 one second (100m sampling lines) plus 1 sec nadir dwell, has a larger representativeness error than a system with (5)13 second lines….unless there are few to no organized circulations(e.g. OLEs or plumes) on scales of order 1km in the LES simulation. In that case, the advantage seems to be all in having 5 lidars operating simultaneously (5*13 = 65 vs. 12) or a factor of 5 used to beat down the MAE due to random turbulence on the scales of 100m and less.

Answer to 2d: We agree, the reason for the reduced representativeness error is because the five lidars are operating simultaneously. Thereby, different parts of the retrieval volume are constantly explored, whereas the scanning system can only measure at one location at a time. The results are independent of whether a frozen turbulence assumption is used or not. One can also view it from a perspective as mentioned by the reviewer: With five beams, five times the measuring distance is available inside each retrieval volume, compared to a single beam system. However, the reduction in retrieval error is not a full factor of five since the sampling error scales approximately with the root of the number of measurements conducted. Further, the simultaneous measurements from the multiple beams are not fully independent due to the spatial correlation of turbulence and the sampling of coherent eddy structures.